# Long non-coding RNA *Neat1* and paraspeckle components are translational regulators in hypoxia

**Anne-Claire Godet**[1†], **Emilie Roussel**[1†], **Florian David**[1], **Fransky Hantelys**[1], **Florent Morfoisse**[1], **Joffrey Alves**[1], **Françoise Pujol**[1], **Isabelle Ader**[2], **Edouard Bertrand**[3], **Odile Burlet-Schiltz**[4], **Carine Froment**[4], **Anthony K Henras**[5], **Patrice Vitali**[5], **Eric Lacazette**[1], **Florence Tatin**[1], **Barbara Garmy-Susini**[1], **Anne-Catherine Prats**[1]*

[1]UMR 1297-I2MC, Inserm, Université de Toulouse, Toulouse, France; [2]UMR 1301-RESTORE, Inserm, CNRS 5070, Etablissement Français du Sang-Occitanie (EFS), National Veterinary School of Toulouse (ENVT), Université de Toulouse, Toulouse, France; [3]UMR5535 CNRS-IGMM, Université de Montpellier, Montpellier, France; [4]Institut de Pharmacologie et Biologie Structurale (IPBS), Université de Toulouse, CNRS, Toulouse, France; [5]Molecular, Cellular and Developmental Biology Unit (MCD), Centre de Biologie Intégrative (CBI), Université de Toulouse, Toulouse, France

**\*For correspondence:**
anne-catherine.prats@inserm.fr

[†]These authors contributed equally to this work

**SUMMARY** Internal ribosome entry sites (IRESs) drive translation initiation during stress. In response to hypoxia, (lymph)angiogenic factors responsible for tissue revascularization in ischemic diseases are induced by the IRES-dependent mechanism. Here, we searched for IRES *trans*-acting factors (ITAFs) active in early hypoxia in mouse cardiomyocytes. Using knock-down and proteomics approaches, we show a link between a stressed-induced nuclear body, the paraspeckle, and IRES-dependent translation. Furthermore, smiFISH experiments demonstrate the recruitment of IRES-containing mRNA into paraspeckle during hypoxia. Our data reveal that the long non-coding RNA *Neat1*, an essential paraspeckle component, is a key translational regulator, active on IRESs of (lymph)angiogenic and cardioprotective factor mRNAs. In addition, paraspeckle proteins p54[nrb] and PSPC1 as well as nucleolin and RPS2, two p54[nrb]-interacting proteins identified by mass spectrometry, are ITAFs for IRES subgroups. Paraspeckle thus appears as a platform to recruit IRES-containing mRNAs and possibly host IRESome assembly. Polysome PCR array shows that *Neat1* isoforms regulate IRES-dependent translation and, more widely, translation of mRNAs involved in stress response.

## Editor's evaluation

The paper reports that the long non-coding RNA *Neat1* (nuclear paraspeckle assembly transcript 1) is required for IRES Internal Ribosome Entry Site)-mediated mRNA translation activity. *Neat1* is required for the activity of many cellular IRESs during the stress response in angiogenesis and/or cardio-protection. The authors conclude that nuclear paraspeckles serve as areas where cellular IRESes acquire ITAFs (IRES trans-activating factors. The findings of this paper have practical implications beyond a single subfield and the methods, data, and analyses broadly support the claims with only minor weaknesses.

## Introduction

Cell stress triggers major changes in the control of gene expression at the transcriptional and post-transcriptional levels. One of the main responses to stress is the blockade of global translation allowing cells to save energy. This process results from inactivating the canonical cap-dependent mechanism of translation initiation (*Holcik and Sonenberg, 2005*). However, translation of specific mRNAs is maintained or even increased during stress via alternative mechanisms of translation initiation. One of these mechanisms involves internal ribosome entry sites (IRES), structural elements mostly present in the 5′ untranslated regions of specific mRNAs, which drive the internal recruitment of ribosomes onto mRNA and promote cap-independent translation initiation (*Godet et al., 2019*).

Hypoxia, or the lack of oxygen, is a major stress in pathologies such as cancer and cardiovascular diseases (*Pouysségur et al., 2006*). In particular, in ischemic heart failure disease, coronary artery branch occlusion exposes cardiac cells to hypoxic conditions. The cell response to hypoxia induces angiogenesis and lymphangiogenesis to reperfuse the stressed tissue with new vessels and allow cell survival (*Morfoisse et al., 2014*; *Pouysségur et al., 2006*; *Tatin et al., 2017*). The well-known response to hypoxia is the transcriptional induction of specific genes under the control of the hypoxia-induced factors 1 and 2 (HIF1, HIF2) (*Hu et al., 2003*; *Koh et al., 2011*). However, we have recently reported that most mRNAs coding (lymph)angiogenic growth factors are induced at the translatome level in hypoxic cardiomyocytes (*Hantelys et al., 2019*). Expression of these factors allows the recovery of functional blood and lymphatic vasculature in ischemic diseases, including myocardial infarction (*Tatin et al., 2017*; *Ylä-Herttuala and Baker, 2017*). The mRNAs of the major (lymph)angiogenic growth factors belonging to the fibroblast growth factor (FGF) and vascular endothelial growth factor (VEGF) families all contain IRESs that are activated in early hypoxia (*Morfoisse et al., 2014*; *Hantelys et al., 2019*).

IRES-dependent translation is regulated by IRES trans-acting factors (ITAFs) that are in most cases RNA-binding proteins acting as positive or negative regulators. A given ITAF can regulate several IRESs, while a given IRES is often regulated by several ITAFs (*Godet et al., 2019*), depending on the cell type or physiology. This has led to the concept of IRESome, a multi-partner ribonucleic complex allowing ribosome recruitment onto the mRNA via the IRES.

ITAFs often exhibit several functions in addition to their ability to control translation. Many of them play a role in alternative splicing, transcription, ribosome biogenesis or RNA stability (*Godet et al., 2019*). Clearly, a large part of ITAFs are nuclear proteins able to shuttle between nucleus and cytoplasm. Previous data have also shown that a nuclear event is important for cellular IRES activity, leading to the hypothesis of IRESome formation in the nucleus (*Ainaoui et al., 2015*; *Semler and Waterman, 2008*; *Stoneley et al., 2000*).

Interestingly, several ITAFs are components of a nuclear body, the paraspeckle, formed in response to stress (*Choudhry et al., 2015*; *Fox et al., 2002*). These ITAFs include several hnRNPs, as well as major paraspeckle proteins such as P54$^{nrb}$ nuclear RNA binding (P54$^{nrb}$/NONO) and splicing factor proline and glutamine-rich (SFPQ/PSF). P54$^{nrb}$ and SFPQ belong to the family of *Drosophila melanogaster* behavior and human splicing (DBHS) proteins whose third member is the paraspeckle protein C1 (PSPC1). P54$^{nrb}$ and SFPQ are essential for paraspeckle formation while PSPC1 is not. These three DBHS proteins are known to interact with each other and function in heteroduplexes (*Fox et al., 2005*; *Lee et al., 2015*; *Passon et al., 2012*). In addition, P54$^{nrb}$ and SFPQ interact with the long non-coding RNA (lncRNA) *Neat1* (nuclear enriched abundant transcript 1), that constitutes the skeleton of the paraspeckle (*Clemson et al., 2009*; *Sunwoo et al., 2009*). This lncRNA, a paraspeckle essential component, is present as two isoforms *Neat1_1* and *Neat1_2* whose sizes in mouse are 3.2 and 20.8 kilobases, respectively (*Sunwoo et al., 2009*). Its transcription is induced during hypoxia by HIF2 and promotes paraspeckle formation (*Choudhry et al., 2015*). *Neat1* is overexpressed in many cancers (*Yang et al., 2017*). Recently, its induction by hypoxia has been shown in cardiomyocytes where it plays a role in cell survival (*Kenneweg et al., 2019*).

According to previous reports, paraspeckle is able to control gene expression via the retention of edited mRNAs and transcription factors (*Hirose et al., 2014*; *Imamura et al., 2014*; *Prasanth et al., 2005*). In 2017, Shen et al. have also shown that the paraspeckle might inhibit translation by sequestering p54$^{nrb}$ and SFPQ which are ITAFs of the *MYC* IRES (*Shen et al., 2017*).

In this study, we were interested in finding new ITAFs responsible for activating (lymph)angiogenic factor mRNA IRESs in HL-1 cardiomyocytes, during early hypoxia. We have previously shown that

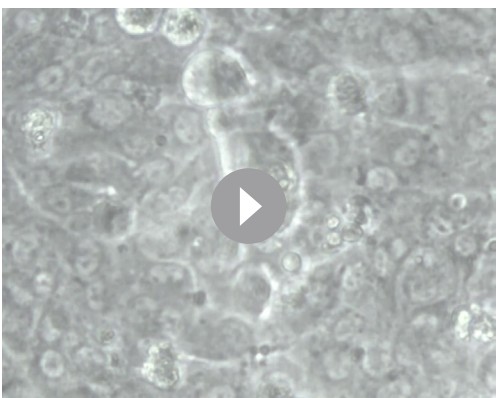

**Video 1.** Beating HL-1 cardiomyocytes (Enlargement 40X). Mouse atrial HL-1 cardiomyocytes exhibit a beating phenotype when cultured in Claycomb medium at high density (*Claycomb et al., 1998*). This phenotype was required to obtain all the data described in the present study.
https://elifesciences.org/articles/69162/figures#video1

the two paraspeckle proteins p54nrb and hnRNPM are ITAFs, activators of the FGF1 IRES during myoblast differentiation (*Ainaoui et al., 2015*). This incited us to investigate the potential role of the paraspeckle and of *Neat1* in the control of IRES-dependent translation in hypoxic cardiomyocytes. We show here that Neat1 expression and paraspeckle formation correlate with the activation of the *FGF1* IRES during hypoxia, in cardiomyocytes and breast cancer cells. The knock-down of p54nrb, PSPC1 or *Neat1* generates a decrease in *FGF1* IRES activity and in endogenous FGF1 expression. Furthermore, our data revealed that IRES-containing mRNA is colocalized with Neat1 in paraspeckle during hypoxia. By quantitative mass spectrometry analysis of the p54nrb interactome, we identified two additional ITAFs able to control the *FGF1* IRES activity: nucleolin and ribosomal protein RPS2. Analysis of IRESs in the knock-down experiments showed that p54nrb and PSPC1 are activators of several but not all IRESs of (lymph)angiogenic and cardioprotective factor mRNAs whereas *Neat1* appears as a strong activator of all the cellular IRESs tested. These data suggest that the paraspeckle, via *Neat1* and several protein components would be the site of IRESome assembly in the nucleus. In addition, a polysome PCR array reveals that *Neat1* affects the translation of most IRES-containing mRNAs and of several mRNA families involved in hypoxic response, angiogenesis and cardioprotection.

## Results

### FGF1 IRES activation during hypoxia correlates with paraspeckle formation and with Neat1 induction in different cell types

In order to analyze the regulation of IRES activity during hypoxia, HL-1 cardiomyocytes were transduced with the 'Lucky Luke' bicistronic lentivector validated in our previous reports, containing the *renilla* luciferase (LucR) and firefly luciferase (LucF) genes separated by the *FGF1* IRES (*Video 1*, *Figure 1A*). In this construct, the first cistron LucR is expressed in a cap-dependent manner and the second cistron LucF is under the control of the IRES. The ratio LucF/LucR reflects the IRES activity.

LucR and LucF activities were measured in HL-1 cells subjected to hypoxia for 4 hr, 8 hr, or 24 hr (*Figure 1—figure supplement 1*, *Supplementary file 1*). These conditions were exactly the same as that used in our previous report providing evidence of IRES activation by hypoxia (*Hantelys et al., 2019*). We previously showed in the same report that eIF2α is phosphorylated after 4 hr of hypoxia, while no change in 4E-BP1 phosphorylation is observed. The polysome/monosome ratio indicated that global protein synthesis decreases in these conditions (*Hantelys et al., 2019*). Those data allowed us to conclude that IRES activities are not negatively affected by eIF2α phosphorylation.

Here, we showed that both luciferase activities increase after 4 hr of hypoxia and decreased at 24 hr. However, LucF increased more than LucR (2.5 times versus 1.5 times, respectively). Thus the ratio LucF/LucR revealed a significant activation of the *FGF1* IRES in early hypoxia, correlated to induction of endogenous FGF1 as previously shown (*Hantelys et al., 2019*; *Figure 1B and C*, *Figure 1—figure supplement 1*). *Neat1* and *Neat1_2* expression in cells was measured by reverse transcription and droplet digital PCR (RT ddPCR), showing an increase of *Neat1* and *Neat1_2* at 4 hr with a peak of expression of Neat1 at 8 hr of hypoxia, while the peak of expression of *Neat1_2* was observed after 4 hr of hypoxia (*Figure 1D*). The same data were also obtained by classical RT-qPCR (data not shown), in agreement with our previous report showing *Neat1* induction by hypoxia in HL-1 cells (*Hantelys et al., 2019*).

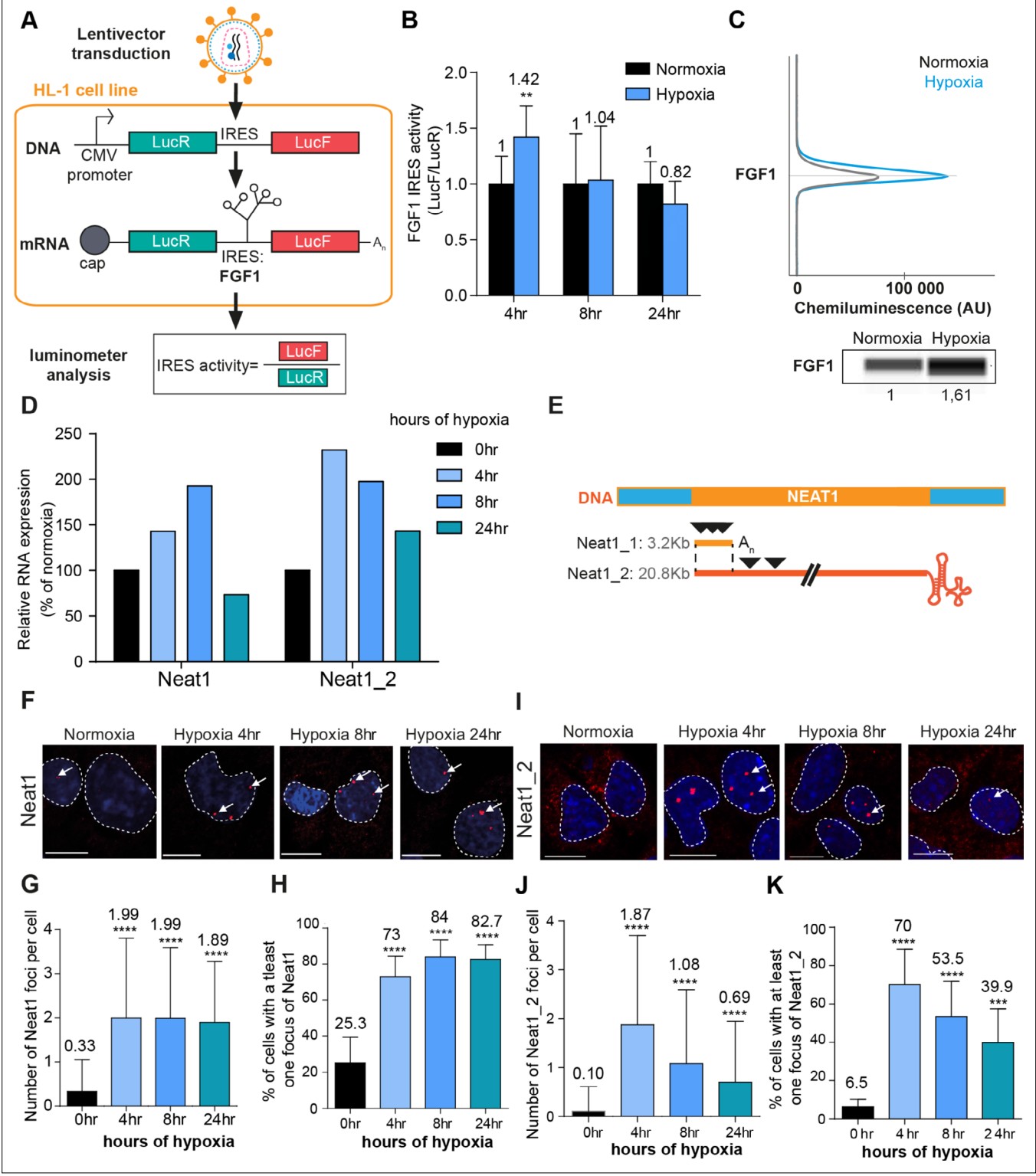

**Figure 1.** *FGF1* IRES activation during hypoxia correlates with *Neat1* induction and paraspeckle formation. (**A**) Schema depicting the Lucky Luke bicistronic construct and HL-1 cells transduced by a lentivector carrying the transgene. The LucF/LucR ratio indicates the IRES activity. (**B**) Activity of the human *FGF1* IRES in HL-1 cardiomyocytes at 4 hr, 8 hr, or 24 hr of hypoxia normalized to normoxia. The corresponding luciferase values are presented in *Figure 1—figure supplement 1*, *Supplementary file 1*. (**C**) Detection of endogenous mouse FGF1 by capillary Simple Western in normoxic and hypoxic (2 hr) cardiomyocytes. The curve corresponds to the chemiluminescence signal detected with FGF1 antibody. A numerical blot is represented. Below the blot is shown the quantification of FGF1 normalized to total proteins and to control gapmer. Total proteins are detected by a dedicated

*Figure 1 continued on next page*

*Figure 1 continued*

channel in capillary Simple Western. The full raw unedited gel is provided in *Figure 1—figure supplement 1* (*Figure 1—figure supplement 1—source data 1*). (**D**) HL-1 cells were subjected to normoxia (0 hr) or to hypoxia during 4 hr, 8 hr, and 24 hr. *Neat1* and *Neat1_2* expression was analyzed by droplet digital PCR (Primer sequences in *Supplementary file 2*). RNA expression is normalized to the normoxia time point. (**E**) Schema depicting the *Neat1* mouse gene and the *Neat1_1* and *Neat1_2* RNA isoform carrying a poly(A) tail or a triple helix, respectively. Black arrowheads represent FISH probes against *Neat1* and *Neat1_2* (sequences in *Supplementary file 2*). (**F–K**) *Neat1* (**F**) or *Neat1_2* (**I**) FISH labeling in HL-1 cardiomyocytes in normoxia or at 4 hr, 8 hr, and 24 hr of 1% $O_2$. DAPI staining is represented in blue and *Neat1* or *Neat1_2* cy3 labeling in red. Nuclei are delimited by dotted lines. Scale bar = 10 μm. Larger fields are presented in *Figure 1—figure supplement 2*. (**G and J**) Quantification of *Neat1* (**G**) or *Neat1_2* (**J**) foci per cell by automated counting (ImageJ). (**H and K**) Percentage of cell harboring at least one focus of *Neat1* (**H**) or *Neat1_2* (**K**); Histograms correspond to means ± standard deviation, with Mann-Whitney (n=12) (**B**) or one-way ANOVA (**G-H**, n=269–453) and (**J-K**, n=342–499); **p<0.01, ***<0.001, ****p<0.0001.

The online version of this article includes the following source data and figure supplement(s) for figure 1:

**Figure supplement 1.** Bicistronic vector LucR and LucF expression and endogenous FGF1 protein expression in hypoxic HL-1 cells.

**Figure supplement 1—source data 1.**

**Figure supplement 2.** Detection of *Neat1* and *Neat1_2* in hypoxic HL-1 by FISH.

**Figure supplement 3.** *FGF1* IRES is activated by hypoxia in correlation with *Neat1* induction in 67NR cells and inactivated after *Neat1* knock-down.

In parallel, paraspeckle formation was studied by fluorescent in situ hybridization (FISH) targeting the non-coding RNA *Neat1*, considered as the main marker of paraspeckles. The fluorescent probes targeted either the common part of the two isoforms *Neat1_1* and *Neat 1_2*, or only the large isoform *Neat1_2* (*Figure 1E*). After 4 hr of hypoxia, the number of foci increased and reached 2 foci per cell on average, while the number of cells containing at least one focus shifted from 20% to 70% (*Figure 1F–K*, *Figure 1—figure supplement 2*). This was observed with both *Neat1* and *Neat1_2* probes. The values observed at 4 hr did not change after 8 hr and 24 hr of hypoxia with the *Neat1* probe (*Figure 1F–H*). In contrast, the number of foci containing *Neat1_2* decreased after longer times of hypoxia: at 8 hr and 24 hr, the number of foci per cell reached 1 and 0.5 while only 50% and 40% of the cells contained at least one focus, respectively (*Figure 1I–K*). Surprisingly, *Neat1_2* was detected in the cytoplasm in normoxia and after 24 hr of hypoxia (*Figure 1I*, *Figure 1—figure supplement 2*).

These data revealed that *FGF1* IRES activation correlates with increased *Neat1* expression and paraspeckle formation after 4 hr of hypoxia in HL-1 cardiomyocytes. To determine whether such a correlation also occurs in other cell types, similar experiments were performed in a mouse breast tumor cell line 67NR (*Figure 1—figure supplement 3*). In these cells, known to be more resistant to hypoxia, *Neat1* increased only after 24 hr of hypoxia. In particular, we observed a strong and significant induction of *Neat1_2* (*Figure 1—figure supplement 3B*). As regards the IRES activity (LucF/LucR ratio), it also increased after 24 hr of hypoxia (*Figure 1—figure supplement 3C*).

These data indicate that the correlation between *Neat1_2* isoform induction and IRES activation under hypoxia exists in different cell types.

## LncRNA *Neat1* knock-down drastically affects the *FGF1* IRES activity and endogenous FGF1 expression

To determine whether *Neat1* could have a role in the regulation of *FGF1* IRES activity, we depleted HL-1 for this non-coding RNA using locked nucleic acid (LNA) gapmers, antisense modified oligonucleotides described for their efficiency in knocking-down nuclear RNAs. HL-1 cells transduced with the bicistronic vector were transfected with a pool of gapmers targeting *Neat1* and with a control gapmer (*Supplementary file 2*). The knock-down efficiency was measured by smiFISH (single molecule inexpensive FISH) and ddPCR and showed a decrease in the number of paraspeckles, correlated to the decrease of *Neat1* RNA, which shifted from 5 to 2 foci per cell (*Figure 2A–B*, *Figure 2—figure supplement 1A*; *Tsanov et al., 2016*). In these experiments performed in normoxia, the number of paraspeckles was high (almost 5 foci per cell), suggesting that cells were already stressed by the gapmer treatment, before being submitted to hypoxia. Alternatively, it could also be explained by the high sensitivity of the smiFISH method used here, whereas paraspeckles were detected by FISH in *Figure 1*. To evaluate the IRES activity, the ratio LucF/LucR was measured in normoxia or after 4 hr of hypoxia, revealing that the IRES activity decreased by two times upon Neat1 depletion (*Figure 2C*, *Supplementary file 3*). This effect was also observed on endogenous FGF1 protein expression,

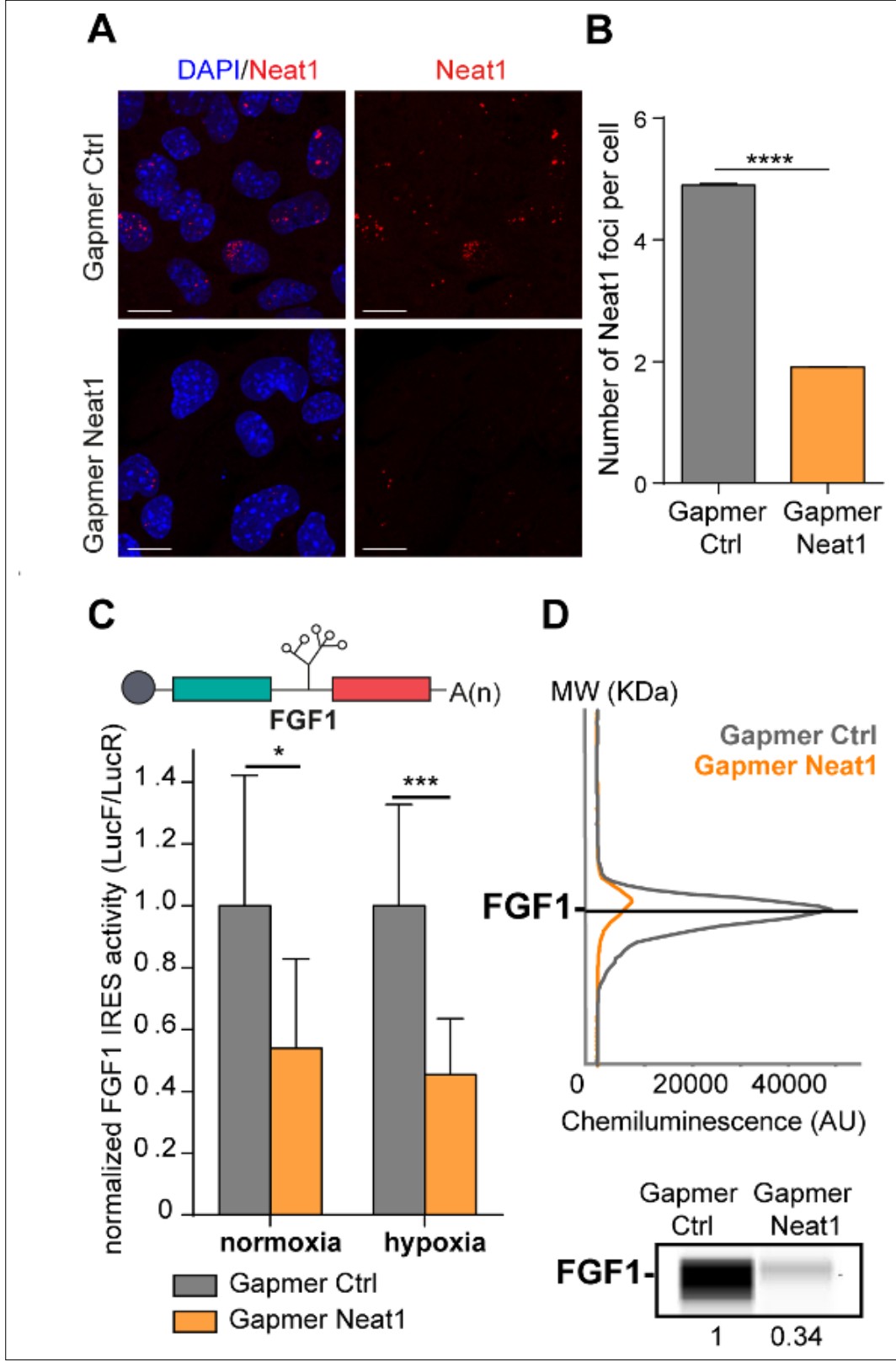

**Figure 2.** LncRNA *Neat1* knock-down drastically affects the *FGF1* IRES activity and endogenous FGF1 expression. (**A**) SmiFISH imaging of *Neat1* knock-down by a pool of LNA gapmers targeting both isoforms (Sequences in *Supplementary file 2C*). Cells were treated during 48 hr with the gapmers. Scale bar = 10 μm. (**B**) *Neat1* foci counting per cell for the control gapmer and *Neat1* LNA gapmer pool, using unpaired two-tailed student t-test

*Figure 2 continued on next page*

*Figure 2 continued*

with n=249 for control and 187 for Neat1 LNA gapmer. (**C**) *FGF1* IRES activities in HL-1 cells transduced with Lucky Luke bicistronic reporter and treated with gapmer *Neat1* or control during normoxia or hypoxia (1% O$_2$). Histograms correspond to means ± standard deviation of the mean. Non-parametric Mann-Whitney test was performed with n=9. *p<0.05, ***<0.001, ****p<0.0001. The mean has been calculated with nine cell culture biological replicates, each of them being already the mean of three technical replicates (27 technical replicates in total). Detailed values of biological replicates are presented in **Supplementary file 3**. (**D**) Detection of endogenous mouse FGF1 by capillary Simple Western. The curve corresponds to the chemiluminescence signal detected with FGF1 antibody. A numerical blot is represented. Below the blot is shown the quantification of FGF1 normalized to total proteins and to control gapmer. The source data of the capillary Simple Western are provided in **Figure 2—figure supplement 2**. Total proteins are detected by a dedicated channel in capillary Simple Western.

The online version of this article includes the following source data and figure supplement(s) for figure 2:

**Figure supplement 1.** Knock-down of *Neat1* and *Neat1_2* in HL-1 cardiomyocytes.

**Figure supplement 2.** Effect of *Neat1* knock-down on endogenous FGF1 protein expression.

**Figure supplement 2—source data 1.**

**Figure supplement 3.** Effect of *Neat 1_2* knock-down on *FGF1* IRES activity.

**Figure supplement 4.** Effect of *Neat1_2* knock-down on eIF2α phosphorylation.

**Figure supplement 4—source data 1.**

**Figure supplement 5.** FGF1 half-life after *Neat1_2* gapmer treatment.

**Figure supplement 5—source data 1.**

**Figure supplement 6.** p21 half-life after *Neat1_2* gapmer treatment.

**Figure supplement 6—source data 1.**

measured by capillary Simple Western, which decreased by three times (**Figure 2D**, **Figure 2—figure supplement 2**).

*Neat1_2* knock-down was then performed to evaluate the contribution of the long Neat1 isoform. Also, the *FGF1* IRES activity decreased following *Neat1_2* depletion, however less importantly than with the knock-down of the two isoforms (**Figure 2—figure supplement 3**), suggesting an involvement of both *Neat1* isoforms. Capillary Western experiments indicated a slight increase of eIF2α phosphorylation upon *Neat1_2* depletion (**Figure 2—figure supplement 4**). It was not sufficient to block global translation, as shown by the renilla luciferase activity (**Supplementary file 3**, page 2). Furthermore, we have shown in a previous report that the FGF1 IRES activity increases in hypoxia in conditions of strong eIF2α phosphorylation. FGF1 half-life was superior to 24 hr and was not affected by *Neat1* knock-down (**Figure 2—figure supplements 5–6**). All these arguments indicate that the significant decrease of FGF1 IRES activity and of endogenous FGF1 expression observed in **Figure 2** does not result from eIF2α phosphorylation or decrease in FGF1 half-life, and probably results from Neat1 depletion. This suggested that *Neat1* might regulate *FGF1* mRNA translation, directly or indirectly.

## The IRES-containing mRNA is colocalized with *Neat1* during hypoxia

The effect of *Neat1* on *FGF1* IRES activity suggested an interaction (direct or indirect) between these two RNAs. SmiFISH experiments were performed with two sets of 48 primary probes targeting *Neat1* or the bicistronic mRNA, respectively. As a control, we also used a bicistronic construct with a hairpin instead of the IRES. The two secondary probes were coupled to different fluorophores to detect *Neat1* and the bicistronic mRNA separately and look for a putative colocalization (**Figure 3**). Data clearly show that the IRES containing bicistronic mRNA is colocalized with *Neat1* and that this colocalization significantly increases during hypoxia, which is not the case for the hairpin control (**Figure 3C and D**). These data suggested that the IRES-containing mRNA is recruited into paraspeckles during hypoxia.

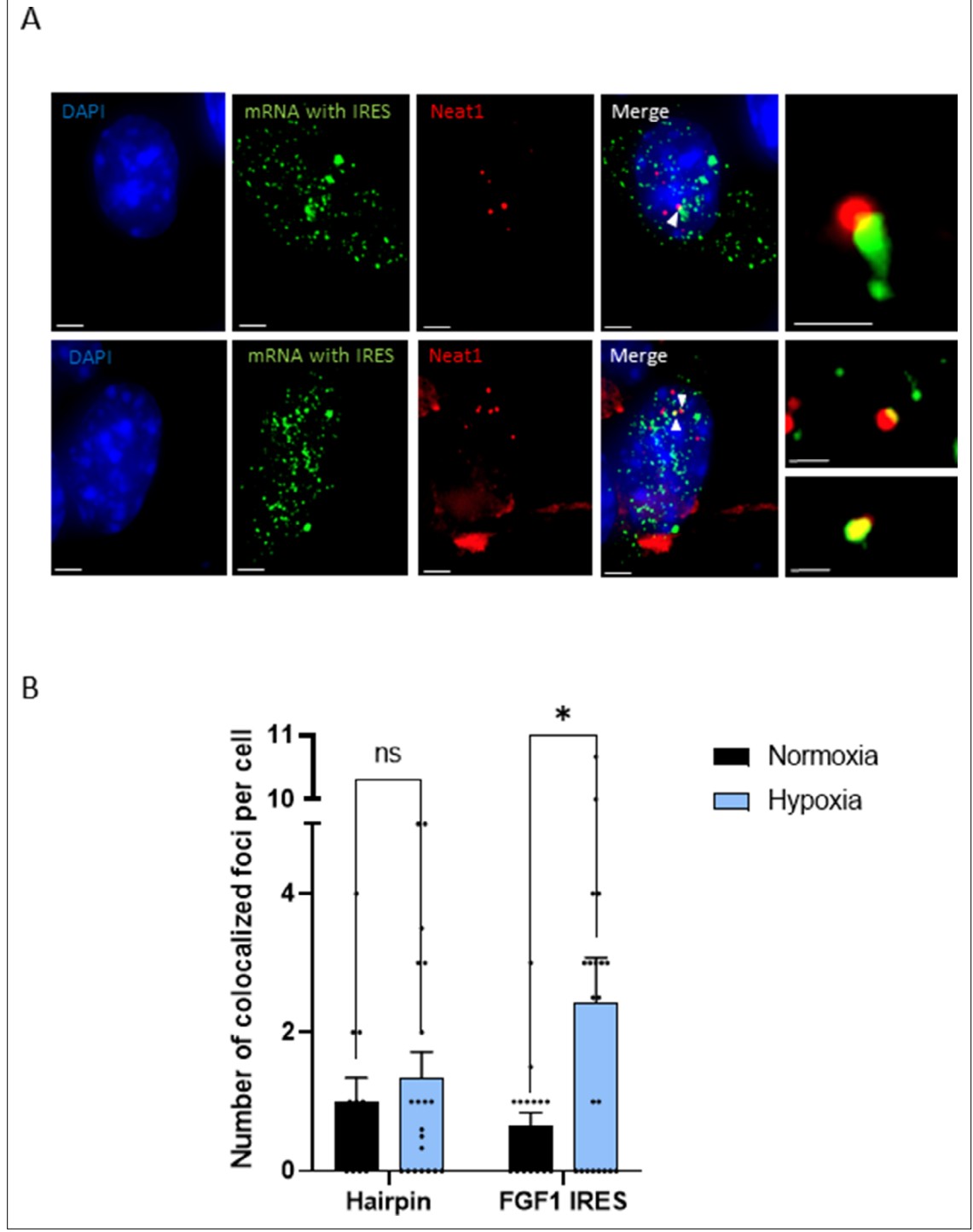

**Figure 3.** IRES-containing mRNA is colocalized with *Neat1* in hypoxic HL-1 cells. Cells were transduced with lentivectors carrying bicistronic Lucky Luke constructs with the *FGF1* IRES or a hairpin (control), subjected or not to 4 hr hypoxia. SmiFISH experiments were performed. (**A**) SmiFISH images showing the bicistronic mRNA carrying the *FGF1* IRES (green) colocalized with *Neat1* RNA (red) in hypoxia condition. Two representative cells are presented. Scale bars are 3 μm for higher panels, 4 μm for lower panesl and 1 μm for zoomed images of colocalized spots. (**B**) Quantification of colocalized spots per cell (n=30). Unpaired two-tailed Student T-test was performed.

## Paraspeckle proteins P54[nrb] and PSCP1, but not SFPQ, are ITAFs of the *FGF1* IRES

The correlation between paraspeckle formation and *FGF1* IRES activation, together with the probable recruitment of IRES-containing mRNA into paraspeckles during hypoxia, incited us to study the role of other paraspeckle components in the control of IRES activity. Three major paraspeckle proteins were chosen, the DBHS proteins, SFPQ, p54[nrb] and PSPC1 (*Figure 4A*). SFPQ and p54[nrb] have been previously described for their ITAF function (*Ainaoui et al., 2015*; *Cobbold et al., 2008*; *Lampe et al., 2018*; *Sharathchandra et al., 2012*; *Shen et al., 2017*). In particular, p54[nrb] regulates the FGF1 *IRES* activity during myoblast differentiation (*Ainaoui et al., 2015*).

HL-1 cells transduced by the 'Lucky Luke' bicistronic construct were transfected with siRNA smart-pools targeting each of the three proteins. The knock-down efficiency was checked by capillary Simple Western, classical Western, or RT qPCR (*Figure 4—figure supplement 1*).

SFPQ knock-down did not affect the IRES activity (*Figure 4B*, *Supplementary file 4*). In contrast, we observed a decrease in IRES activity with p54[nrb] and PSPC1 knock-down, both in normoxia and in hypoxia (*Figure 4C–DSupplementary file 4*, *Supplementary file 5*), despite a knock-down efficiency below 50%. p54[nrb] and PSPC1 knock-down also inhibited the expression of endogenous FGF1 protein (*Figure 3E–F*, *Figure 4—figure supplement 2*). FGF1 half-life was not altered by siRNA treatment, indicating a translational control (*Figure 4—figure supplements 3–4*). These data confirmed the ITAF role of p54[nrb] in HL-1 cardiomyocyte, and indicated that PSPC1 is also an ITAF of the *FGF1* IRES. The ability of three paraspeckle components, *Neat1*, p54[nrb] and PSPC1, to regulate the *FGF1* IRES activity, together with the colocalization of the bicistronic mRNA with Neat1 observed in *Figure 3*, led us to the hypothesis that the paraspeckle might be involved in the control of IRES-dependent translation.

## P54[nrb] interactome in normoxic and hypoxic cardiomyocytes

The moderate effect of p54[nrb] or PSPC1 depletion on *FGF1* IRES activity, possibly due to the poor efficiency of knock-down (>50%), also suggested that other proteins may be involved. Previous data from the literature support the hypothesis that the IRESome is a multi-partner complex. In order to identify other members of this complex, we analysed the p54[nrb] interactome in HL-1 cell nucleus and cytoplasm using a label-free quantitative mass spectrometry approach. For this purpose, cells were transduced by a lentivector expressing an HA-tagged p54[nrb] (*Figure 5A*). After cell fractionation (*Figure 5B* and *Figure 5—figure supplement 1A and B*), protein complexes from normoxic and hypoxic cells were immunoprecipitated with anti-HA antibody. Immunoprecipitated interacting proteins (three to four biological replicates for each group) were isolated by SDS-PAGE, in-gel digested with trypsin and analyzed by nano-liquid chromatography-tandem mass spectrometry (nanoLC-MS/MS), leading to the identification and quantification of 2013 proteins (*Supplementary file 7*). To evaluate p54[nrb] interaction changes, pairwise comparisons based on MS intensity values were performed for each quantified protein between the four groups, cytoplasmic and nuclear complexes from cells subjected to normoxia or hypoxia (*Figure 5C*). Enriched proteins were selected based on their significant protein abundance variations between the two compared group (fold-change (FC) >2 and<0.5, and Student t test p<0.05) (see STAR Method for details) (*Figure 5D–E* and *Figure 5—figure supplement 1*). Globally, the HA-tag capture revealed an enrichment of hnRNP proteins in nucleus and of ribosomal proteins in the cytoplasm (*Figure 5—figure supplement 1C and D*). In nucleus P54[nrb] interacted with itself (endogenous mouse Nono), PSPC1 and SFPQ, as well as with other paraspeckle components: in total P54[nrb] interaction was identified with 22 proteins among 40 paraspeckle components listed in previous reports (*Table 1*; *Naganuma et al., 2012*; *Yamamoto et al., 2021*). Six of these paraspeckle components exhibit an ITAF function (FUS, hnRNPA1, hnRNPK, hnRNPM, hnRNPR, and SFPQ *Figure 5—figure supplement 1*, *Table 1*). Two additional ITAFs interact with p54: hnRNPC and hnRNPI (*Godet et al., 2019*).

As regards cytoplasmic proteins, we identified RPS25, a ribosomal protein previously described as an ITAF for many IRESs (*Figure 5—figure supplement 1A*; *Hertz et al., 2013*). Interestingly, p54[nrb] also interacted with RPS5, RPS18 and RPS19, and other RPs, mainly from the small ribosomal subunit.

Only few proteins were significantly enriched when comparing hypoxic versus normoxic extracts. In hypoxic nucleus, the significantly enriched proteins are hnRNPM, nucleolin (both previously described as ITAFs) (*Hertz et al., 2013*; *Shi et al., 2016*; *Shi et al., 2017*) and the ribosomal protein RPS2/uS5 (*Figure 5D*), while the helicase DDX17, the enolase ENO3 and the heat shock protein HSPA2 are

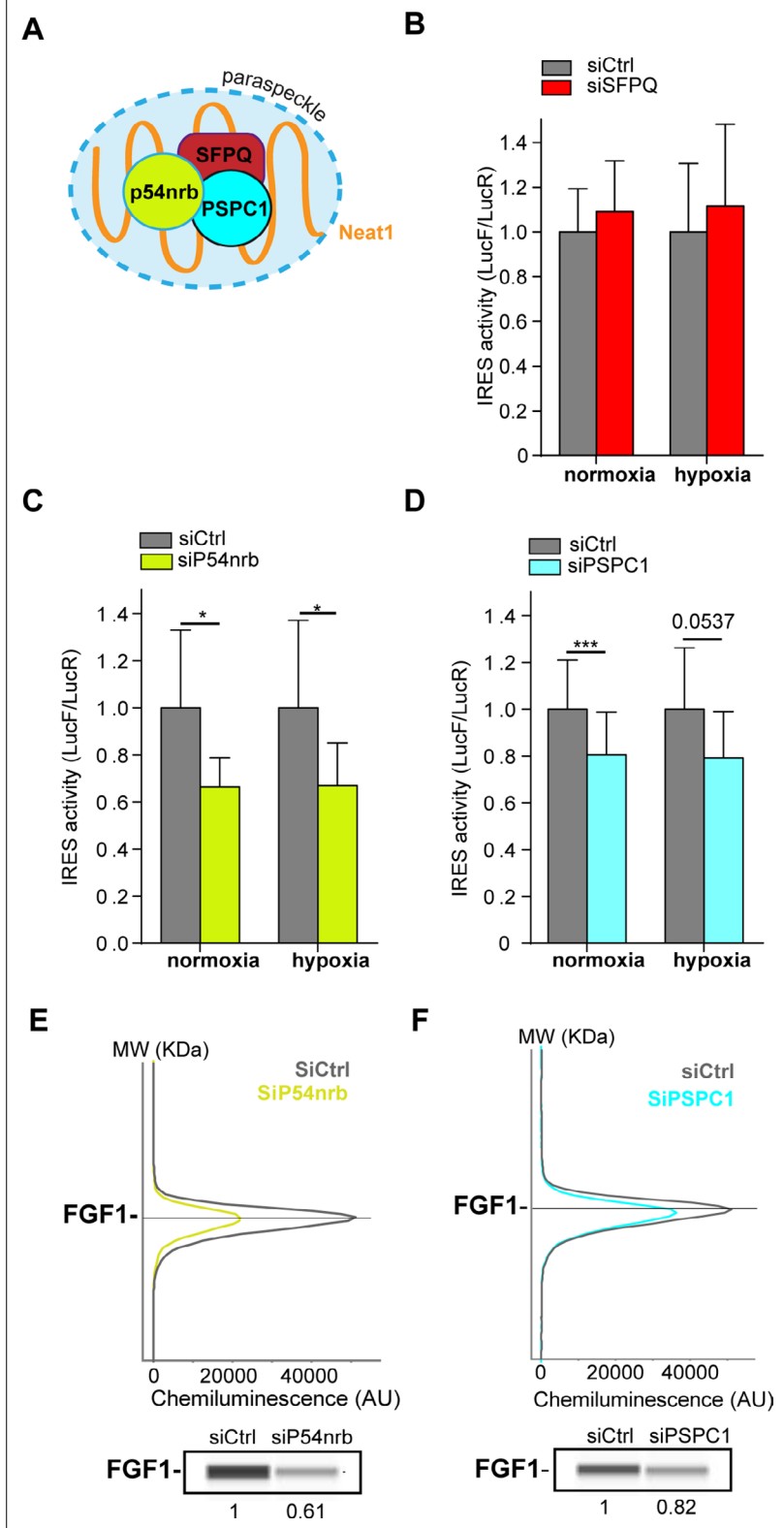

**Figure 4.** Paraspeckle proteins p54[nrb] and PSCP1, but not SFPQ, are ITAFs of the *FGF1* IRES. (**A**) Schema of paraspeckle and DBHS proteins. (**B–D**) *FGF1* IRES activity upon knock-down of SFPQ (**B**), P54[nrb] (**C**) or PSPC1 (**D**) in HL-1 cell (*Figure 4—figure supplement 1—source data 1*) transduced with Lucky Luke bicistronic reporter during normoxia or hypoxia was measured as in *Figure 2*. Cells were harvested 72 hr after siRNA treatment. The IRES activity values have been normalized to the control siRNA. Histograms correspond to means ± standard

*Figure 4 continued on next page*

*Figure 4 continued*

deviation of the mean, with a non-parametric Mann-Whitney test with n=9; *p<0.05, ***<0.001. The mean has been calculated with nine cell culture biological replicates, each of them being already the mean of three technical replicates (27 technical replicates in total). Detailed values of biological replicates are presented in *Supplementary file 3*, *Supplementary file 4*, *Supplementary file 5*. (**E and F**) Capillary Simple Western detection of endogenous FGF1 protein with P54[nrb] (**E**) or PSPC1 (**F**) knock-down. Source data of capillary Simple Western are presented in *Figure 4—figure supplement 2* (*Figure 4—figure supplement 2—source data 1*).

The online version of this article includes the following source data and figure supplement(s) for figure 4:

**Figure supplement 1.** Knock-down of p54[nrb], PCPC1 and SFPQ in HL-1 cardiomyocytes.

**Figure supplement 1—source data 1.**

**Figure supplement 2.** FGF1 protein expression in response to p54[nrb] or PSPC1 knock-down.

**Figure supplement 2—source data 1.**

**Figure supplement 3.** FGF1 half-life in response to p54[nrb] or PSPC1 knock-down.

**Figure supplement 3—source data 1.**

**Figure supplement 4.** p21 half-life in response to p54[nrb] or PSPC1 knock-down.

**Figure supplement 4—source data 1.**

enriched in hypoxic cytoplasm (*Figure 5E*). Interaction of nucleolin with p54[nrb] was also validated by co-immunoprecipitation (*Figure 5—figure supplement 2*).

These data showed that p54[nrb] interacts in normoxia and hypoxia with several ITAFs known as paraspeckle components, suggesting that the paraspeckle might be involved in the formation of the IRESome. Its interaction with numerous RPs also suggests that it interacts with the small ribosomal subunit in the cytoplasm.

## p54[nrb]-interacting proteins, nucleolin and RPS2, control the *FGF1* IRES activity

The three candidates identified in nuclear extracts of hypoxic cardiomyocytes, hnRNPM, nucleolin and RPS2 represent potential candidates as ITAFs of the *FGF1* IRES in hypoxia. Among them, hnRNPM has been previously described as an ITAF during myoblast differentiation while nucleolin is an ITAF of several IRESs including *p53* and *VEGFD* IRESs but has never been described for *FGF1* IRES (*Ainaoui et al., 2015*; *Chen et al., 2012*; *Godet et al., 2019*; *Morfoisse et al., 2016*; *Peddigari et al., 2013*; *Takagi et al., 2005*).

HL-1 cardiomyocytes transduced by the Lucky Luke lentivector with the *FGF1* IRES were transfected as above with siRNA smartpools targeting RPS2, hnRNPM or nucleolin (*Figure 6*). The knock-down was effective, but only 50–60%, for the three mRNAs (*Figure 6A–D*). This moderate knock-down was probably due to a weak transfection efficiency of HL-1 cells with the siRNAs. Nevertheless, we observed a decrease in IRES activity upon depletion of RPS2 and nucleolin, significant in normoxia but with the same trend in hypoxia while no effect was observed upon hnRNPM depletion (*Figure 6E*, *Supplementary file 4*). Nucleolin depletion inhibited endogenous FGF1 protein expression (*Figure 6F*, *Figure 6—figure supplement 1*). These data suggest that nucleolin and RPS2 are new ITAFs of the *FGF1* IRES. Their nuclear localization and interaction with p54[nrb] indicate that they could be components of the paraspeckle. RPS2 has never been described as an ITAF before the present study.

## *Neat1* is the key activator of (lymph)angiogenic and cardioprotective factor mRNA IRESs

We have shown above that three main paraspeckle components, *Neat1*, p54[nrb] and PSPC1, control the FGF1 IRES activity in HL-1 cardiomyocytes. To determine if the role of paraspeckle in translational control may be generalized to other IRESs, we used Lucky Luke lentivectors containing a set of other IRESs from *FGF2, VEGFA, VEGFC, VEGFD, or MYC* genes and from EMCV virus, between the two luciferase genes (*Figure 7*). The *VEGFA* mRNA contains two IRESs called here *VEGFAa* and *VEGFAb* IRESs (*Huez et al., 1998*).

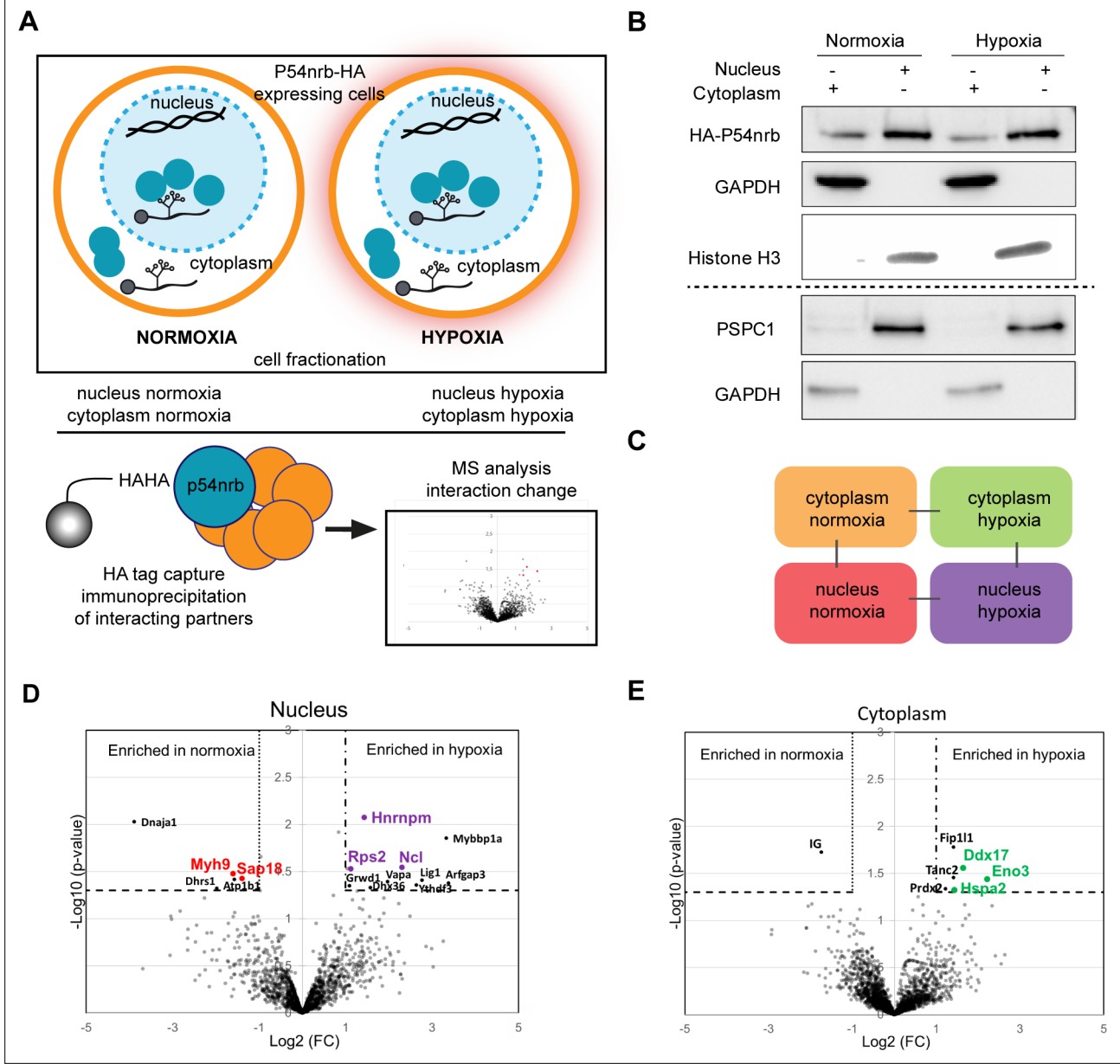

Figure 5. P54nrb interactome in normoxic and hypoxic cardiomyocytes. (A) Experimental workflow: p54nrb-HA transduced HL-1 cells were subjected to normoxia or hypoxia, then nucleus and cytoplasm fractionation was performed and extracts were immunoprecipitated using anti-HA antibody. Enriched interacting proteins were identified by using a label-free quantitative mass spectrometry approach. (B) Western blot of fractionation experiment of HL-1 cells in normoxia and hypoxia. Histone H3 was used as a nuclear control and GAPDH as a cytoplasm control. The dotted line delineates two different blots of the same fractionation experiment. (C) Schema of the four pairwise comparisons submitted to statistical analysis. (D and E) Volcano plots showing proteins enriched (bold black) and significantly enriched (after elimination of false-positive hits from quantitation of low-intensity signals) in the nucleus for hypoxia (purple) versus normoxia (red) (D) or in the cytoplasm for hypoxia (green) versus normoxia (E). An unpaired bilateral student t-test with equal variance was used. Enrichment significance thresholds are represented by an absolute log2-transformed fold-change (FC) greater than 1 and a -log10-transformed (p-value) greater than 1.3. Details are provided in *Supplementary file 7*.

The online version of this article includes the following source data and figure supplement(s) for figure 5:

**Figure supplement 1.** Western blot of fractionation experiment of HL-1 cells and label-free quantitative analysis of HA-P54nrb-bound proteins identified by mass spectrometry in different conditions.

**Figure supplement 2.** p54nrb is co-immunoprecipitated by anti-nucleolin antibody.

**Figure supplement 2—source data 1.**

**Table 1.** The p54 interactome includes 22 among 40 proteins described as paraspeckle components. The paraspeckle components listed in the reports by *Naganuma et al., 2012* and by *Yamamoto et al., 2021* is presented here with their ITAF function and their presence in the p54$^{nrb}$ interactome. Their belonging to class I, II, or III of the paraspeckle proteins is indicated. Class I proteins are essential for paraspeckle formation.

| Name | Alternative name | Class | ITAF | Presence in p54$^{nrb}$ MS-IP |
|---|---|---|---|---|
| ASXL1 | MDS/BOPS | I | No | No |
| CELF6 | | n/d | No | No |
| CIRBP | | IIIB | No | Yes |
| CPSF6 | | IIIA | No | Yes |
| CPSF7 | | II | No | Yes |
| DAZAP1 | | IB | No | Yes |
| DLX3 | | n/d | No | No |
| EWSR1 | | | No | Yes |
| FAM113A | | II | No | No |
| FAM98A | | II | No | Yes |
| FIGN | | II | No | No |
| FUS | | IB | Yes | Yes |
| FUSPI1 | SRSF10 | II | No | Yes |
| hnRNPA1 | | II | Yes | Yes |
| hnRNPA1L2 | | n/d | No | No |
| hnRNPF | | n/d | No | Yes |
| hnRNPH1 | | n/d | No | Yes |
| hnRNPH3 | | IB | No | No |
| hnRNPK | | IA | Yes | Yes |
| hnRNPM | | n/d | Yes | Yes |
| hnRNPR | | II | Yes | No |
| hnRNPUL1 | | II | No | Yes |
| MEX3C | | n/d | No | No |
| NUDT21 | | IIIA | No | Yes |
| p54$^{nrb}$ | NONO | IA | Yes | Yes |
| PSPC1 | | IIIB | No | Yes |
| RBM12 | | II | No | No |
| RBM14 | | IA | No | No |
| RBM3 | | IIIB | No | Yes |
| RBM4B | | IIIB | No | No |
| RBM7 | | IIIB | No | No |
| RBMX | | IIIB | No | Yes |
| RUNX3 | | IIIB | No | No |
| SFPQ | PSF | IA | Yes | Yes |
| SS18L1 | | n/d | No | No |
| SWI/SNF | | IB | No | No |

*Table 1 continued on next page*

Table 1 continued

| Name | Alternative name | Class | ITAF | Presence in p54$^{nrb}$ MS-IP |
|---|---|---|---|---|
| TAF15 | | | No | No |
| TDP-43 | | II | No | No |
| UBAP2L | | IIIA | No | Yes |
| ZNF335 | TARDBP | IIIB | No | Yes |

HL-1 cells were transduced by the different lentivectors and transfected either by the siRNA smart-pools to deplete p54$^{nrb}$ and PSPC1, or by the gapmer pool to deplete *Neat1*. The data revealed that p54$^{nrb}$ or PSPC1 depletion affected several IRESs but not all (*Figure 7A–BSupplementary file 5*, *Supplementary file 6*), whereas *Neat1* depletion clearly affected all cellular IRESs but not the viral EMCV IRES (*Figure 7C*, *Supplementary file 3*).

These data allowed us to group the IRESs in different 'regulons' in normoxia and in hypoxia (*Figure 7D*). According to our data, P54$^{nrb}$ is an activator of the *FGF1* and *VEGFC* IRESs in normoxia, and of the *FGF1* and *VEGFAa* IRESs in hypoxia. PSPC1 is an activator of the *FGF1*, *FGF2*, *VEGFAa*, *VEGFC,* and *IGF1R* IRESs in normoxia and of the *FGF1* and *FGF2* IRESs in hypoxia. *Neat1* is an activator of the *FGF1*, *FGF2*, *VEGFAb*, *VEGFC*, *VEGFD*, *IGF1R,* and *MYC* IRESs but not of the *VEGFAa* IRES in normoxia while it activates all the cellular IRESs in hypoxia. The EMCV IRES does not belong to any of these groups as it is not regulated by these three ITAFs, suggesting that this viral IRES is not regulated by the paraspeckle.

In conclusion, these data suggest that IRESome composition varies for each IRES and with the normoxic or hypoxic conditions. The long non-coding RNA *Neat1* appears as the key ITAF for the activation of all the cellular IRESs, suggesting a crucial role of the paraspeckle in IRESome formation and in the control of IRES-dependent translation, at least for cellular IRESs.

### *Neat1* isoforms impact the recruitment into polysomes of mRNAs involved in the stress response

The role of *Neat1* on translatome was then studied using a Fluidigm Deltagene PCR array targeting 96 genes coding IRES-containing mRNAs, ITAFs or proteins involved in angiogenesis and cardioprotection (*Supplementary file 2E*). HL-1 cells were treated with gapmers targeting the two *Neat1* isoforms or only *Neat1_2* before analyzing the recruitment of mRNAs into polysomes compared to the control gapmer. Recruitment into polysomes decreased for 49% of IRES-containing mRNAs following *Neat1* invalidation, and increased for the other 51%. In contrast this decrease concerned 95% of these mRNAs after *Neat1_2* knock-down (*Figure 8A and B*, *Supplementary file 8*). In contrast, the global level of translation was not affected (*Figure 8—figure supplement 1*). As eIF2α phosphorylation was slightly increased in these conditions (*Figure 2—figure supplement 4*), we cannot completely rule out that it could affect the expression of certain mRNAs, despite the absence of inhibition of global translation. However, the insensitivity of many IRESs to eIF2α phosphorylation shown previously suggests that the present data result from an effect of *Neat1*, particularly on translation of IRES-containing mRNAs, while the two isoforms may have distinct effects (*Hantelys et al., 2019*). Interestingly, a similar effect was observed for the other genes tested in the PCR array: *Neat1* or *Neat1_2* knock-down inhibited translation of ITAF-coding genes by 71% or 87%, respectively (*Figure 8C and D*, *Supplementary file 8*). This inhibition concerned 57% or 89% of the remaining genes involved in angiogenesis and cardioprotection for *Neat1* or *Neat1_2* knock-down, respectively (*Figure 8—figure supplement 2*). In total, 92% of the genes of the PCR array were less recruited into polysomes after *Neat1_2* knock-down, versus only 56% after *Neat1* knock-down. These data strongly suggest that *Neat1_2* might be a translational activator of families of genes involved in the response to hypoxic stress in cardiomyocytes.

## Discussion

The present data demonstrate a link between the paraspeckle and the control of IRES-dependent translation during hypoxia in mouse cardiomyocytes. We show that three major paraspeckle components regulate IRES-dependent translation: p54$^{nrb}$, PSPC1, and *Neat1*, as well as by two proteins

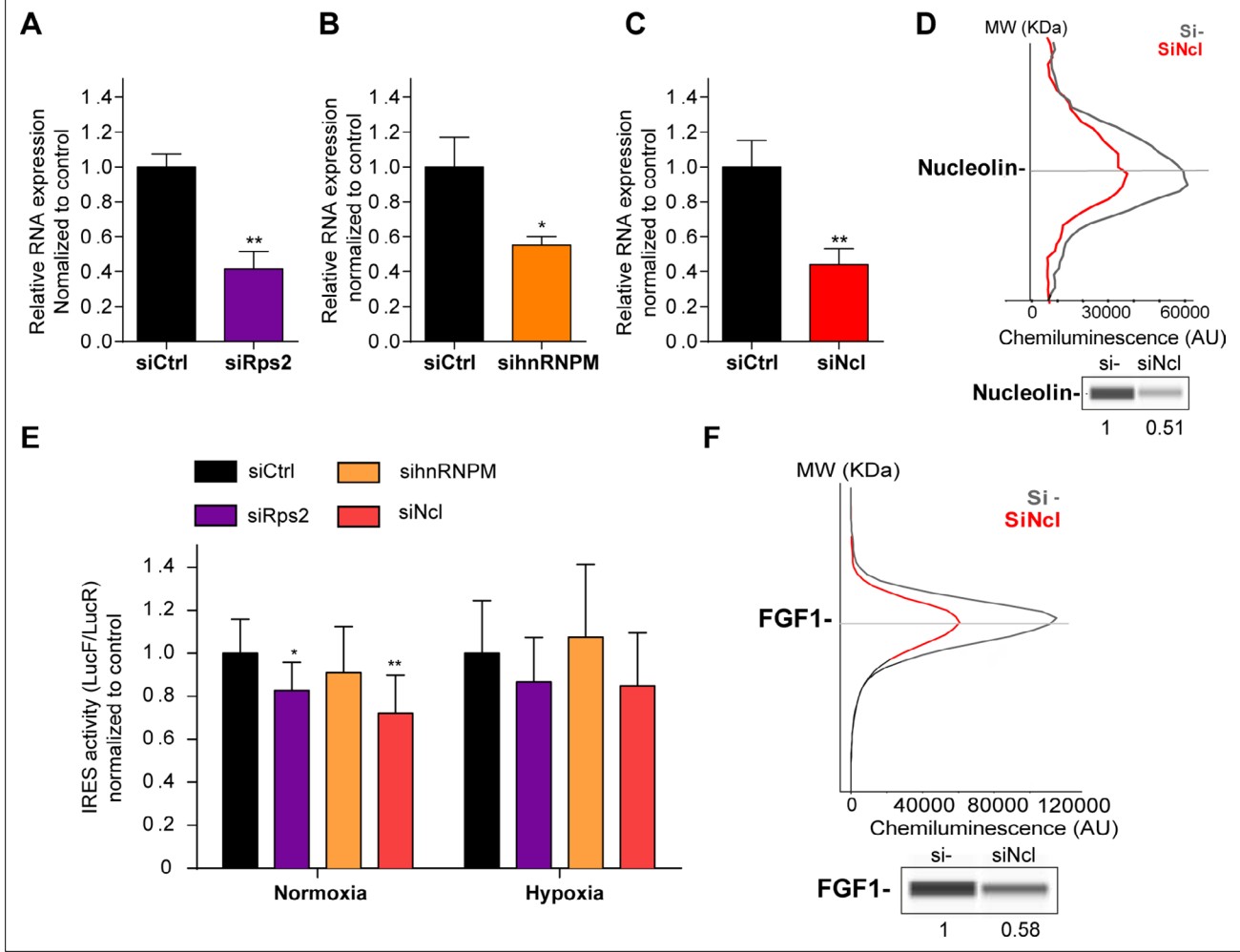

**Figure 6.** p54[nrb]-interacting proteins, nucleolin and RPS2, control the *FGF1* IRES activity. (**A–C**) Quantification of RPS2 (**A**), hnRNPM (**B**) and nucleolin (**C**) RNA expression in HL-1 cells transfected with siRNAs against *Rps2*, hnRNPM or nucleolin mRNA, respectively. RNA expression was measured by RT-qPCR and normalized to control siRNA. One representative experiment is shown with n=3 biological replicates. Student two-tailed t-test was performed with n=3 or Mann-Whitney test with n=9; *p<0.05, **p<0.01, ***<0.001, ****p<0.0001. (**D**) Capillary Simple Western of nucleolin following nucleolin knock-down. The full raw unedited gel is provided in *Figure 6—figure supplement 1A* (*Figure 6—figure supplement 1—source data 1*). (**E**) *FGF1* IRES activity with knock-down by siRNA interference of candidate ITAF nucleolin in HL-1 in normoxia or hypoxia 1% $O_2$ was performed as in *Figure 2*. The IRES activity values have been normalized to the control siRNA. Histograms correspond to means ± standard deviation of the mean, with a non-parametric Mann-Whitney test *p<0.05, **p<0.01. The mean has been calculated with nine cell culture biological replicates, each of them being already the mean of three technical replicates (27 technical replicates in total but the M-W test was performed with n=9). Detailed values of biological replicates are presented in *Supplementary file 6*. (**F**) Capillary Simple Western of endogenous FGF1 following nucleolin knock-down. Histograms correspond to means ± standard deviation. The source data or capillary Simple Western are provided in *Figure 1—figure supplement 1B* (*Figure 6—figure supplement 1—source data 1*).

The online version of this article includes the following source data and figure supplement(s) for figure 6:

**Figure supplement 1.** Endogenous FGF1 protein expression is down-regulated following nucleolin knock-down.

**Figure supplement 1—source data 1.**

present in the p54[nrb] nuclear interactome, nucleolin and RPS2. *Neat1* appears as the key to this paraspeckle-related activation of translation in response to hypoxia. This lncRNA is an activator of all cellular IRESs tested, but not of the viral EMCV IRES. More broadly, *Neat1* isoforms impact the recruitment into polysomes of most IRES-containing mRNAs and several families of mRNAs involved in the response to hypoxia. The colocalization of IRES-containing mRNA with *Neat1* RNA in paraspeckles increased in hypoxia conditions, suggesting that the paraspeckle may be a recruitment platform for

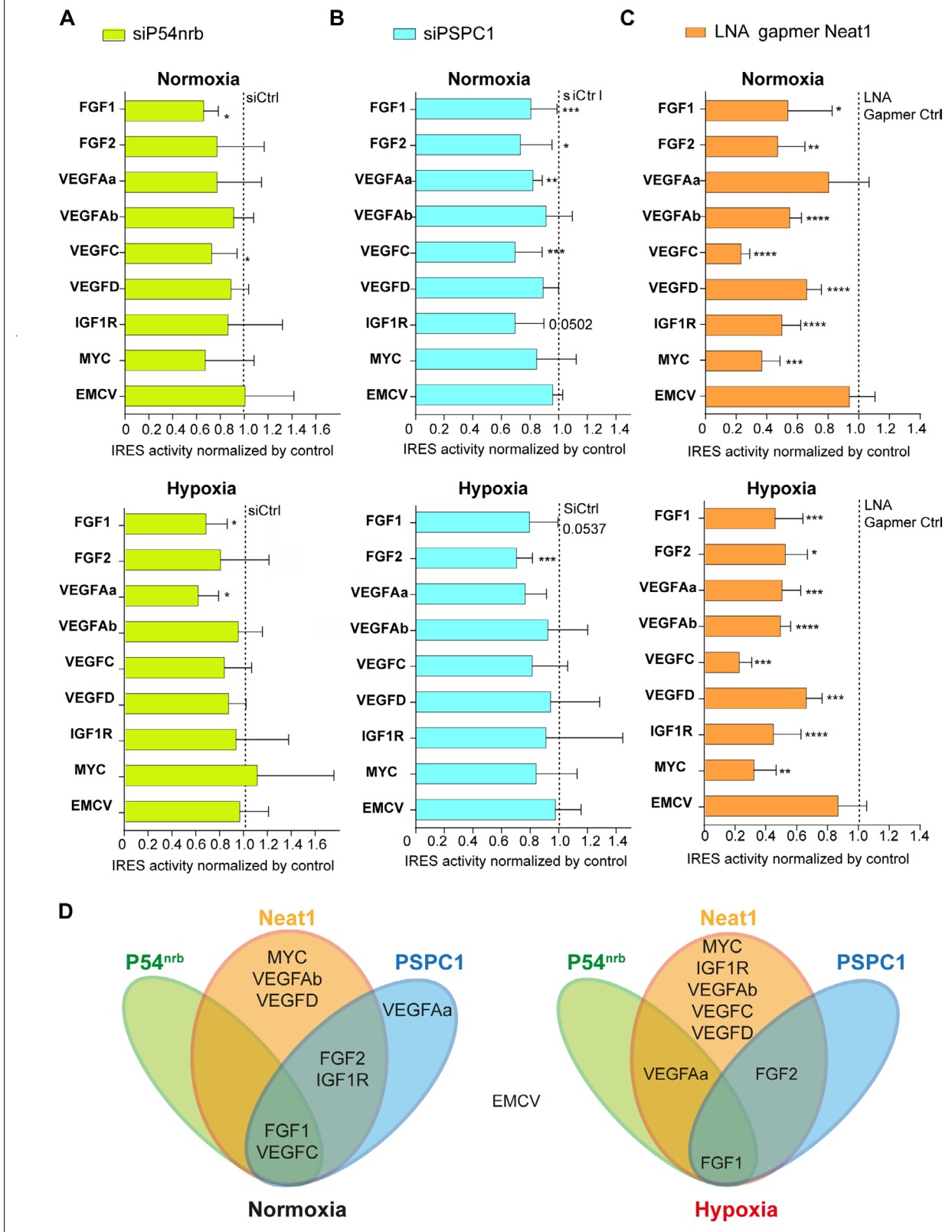

**Figure 7.** *Neat1* is the key activator of (lymph)angiogenic and cardioprotective factor mRNA IRESs. (**A–C**) HL-1 subjected to normoxia or 1% O$_2$ hypoxia were transduced by Lucky Luke bicistronic lentivectors with *FGF1, FGF2, VEGFAa, VEGFAb, VEGFC, VEGFD, IGF1R, MYC,* or EMCV IRES, then the knock-down of p54$^{nrb}$(A) PCPC1 (**B**) and *Neat1* (**C**) was performed as in *Figure 2* and *Figure 4*. IRES activities were measured and normalized to activities in normoxia. IRES activity in normoxia is represented by a dotted line at 1. Histograms correspond to means ± standard deviation, and Mann-Whitney

*Figure 7 continued on next page*

*Figure 7 continued*

test with n=9 or n=12 for *FGF1* IRES; *p<0.05, **p<0.01, ***<0.001, ****p<0.0001. For each IRES the mean has been calculated with nine cell culture biological replicates, each of them being already the mean of three technical replicates (27 technical replicates in total). Detailed values of biological replicates are presented in ***Supplementary file 3***, ***Supplementary file 5***, ***Supplementary file 6***. (**D**) Schema depicting groups of IRESs regulated by *Neat1*, PSPC1, or P54$^{nrb}$ in normoxia or hypoxia.

IRES-containing mRNAs during stress and that the IRESome could be assembled in the paraspeckle before mRNA export from the nucleus (*Figure 9*).

It may be noted that the inhibition of IRES activities resulting from ITAF depletion is quite moderate for the different proteins while stronger for the lncRNA *Neat1*. This cannot be explained only by differences in knock-down efficiency. We hypothesize is that several proteins are present in the IRESome complex and that there may be a certain redundancy between them. Thus, the depletion of a single ITAF would not be sufficient to abolish the IRES activity completely. Also, to explain why paraspeckle ITAFs such as p54$^{nrb}$ and PSPC1 do not inhibit all the IRESs, we propose that the paraspeckle IRESome protein composition varies depending on the IRES and the hypoxic or normoxic condition, while *Neat1* remains the main actor of the process. Several observations suggest that *Neat1_2* may be the main isoform involved. However, the knock-down of *Neat1-2* isoform with a specific gapmer does not affect IRES activity as much as the knock-down of both Neat1 isoforms (*Figure 2—figure supplement 2*). We were not successful in knocking down the isoform Neat1_1, as its sequence is entirely contained in *Neat1_2*. Thus at this stage we conclude that the two isoforms are probably involved. The fluidigm PCR array suggests that they may affect translation differently (*Figure 8*).

We searched for an ITAF able to regulate a set of IRESs during hypoxia and found the lncRNA *Neat1* as a wide activator of IRES-dependent translation. However, our data show that *Neat1* also regulates IRES activities both in normoxia and hypoxia. One explanation may be that *Neat1* is already expressed in normoxia in HL-1 cells, which are transformed cells despite their cardiomyocyte beating phenotype (*Claycomb et al., 1998*). Although *Neat1* expression and paraspeckle number increase in response to hypoxia, a significant percentage of cells already contain paraspeckles in normoxia, which may explain why IRESs are already active in normoxia. It has been reported that *Neat1_2* is not expressed in all tissues in vivo, whereas it is found in all transformed or immortalized cell lines (data not shown) (*Nakagawa et al., 2011*). In concordance with this observation, previous reports show that cellular IRESs are active in all cultured cell lines while inactive or tissue-specific in mice (*Créancier et al., 2000*; *Créancier et al., 2001*). The presence of paraspeckles in normoxia may also reflect the stress due to the transfection procedure, which could interfere with the effect of the hypoxic stress performed in our study. A different approach to obtain *Neat1* silencing, such as CRISPR/Cas9 mediated knock-down or knock-out could provide an interesting solution to this issue.

Our data contrast with the study of Shen et al. who showed that *Neat1* depletion allows redistributing p54$^{nrb}$ and SFPQ/PSF onto the *MYC* mRNA, in correlation with an increase in MYC protein (*Shen et al., 2017*). Several reasons may explain this lack of concordance. Firstly, different cell lines were used: HL-1 cardiomyocytes and 67NR breast tumor cells in the present study, HeLa and MCF7 tumor cells in the report by Shen et al. The regulation of IRES-dependent translation varies depending on cell lines. Secondly, they worked with human cell lines while our report is focused on mouse cells. In human, *MYC* expression is different from mouse as the *MYC* gene contains an additional upstream promoter, P0, which generates a longer transcript with a second IRES (*Nanbru et al., 2001*). Thirdly, they have not directly analyzed the *MYC* IRES activity but only the binding of p54$^{nrb}$ and SFPQ to the *MYC* endogenous mRNA. Moreover an increase in myc protein expression does not necessarily correspond to increased IRES activity as the *MYC* mRNA is also translated by the cap-dependent mechanism (*Nanbru et al., 1997*). Taken together, the two studies are different rather than discordant.

A surprising result has been finding a ribosomal protein, RPS2, in the nuclear p54$^{nrb}$ interactome. This suggests an extra-ribosomal role of this protein. Its interaction with p54$^{nrb}$ favors the hypothesis that RPS2 would impact the IRES activity as an IRESome component in the paraspeckle. The presence of nucleolin in the complex also suggests a link of paraspeckle with nucleolus and ribosome biogenesis. Supporting this, PSPC1 was first identified in the nucleolus proteome (*Fox et al., 2002*). The nuclear binding of specific ribosomal proteins to IRESs might be a mechanism for forming specialized ribosomes.

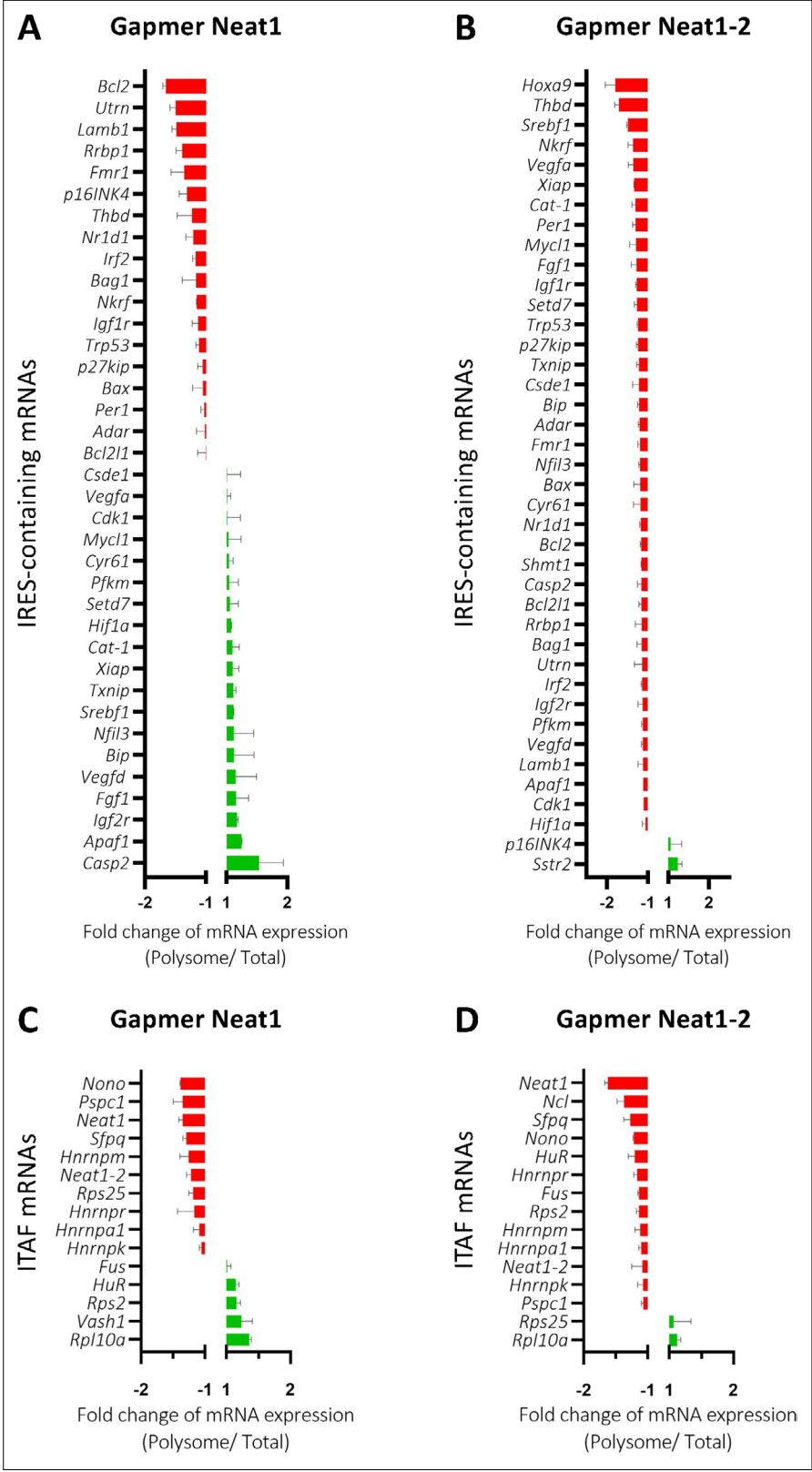

**Figure 8.** *Neat1_2* knock-down down-regulates translation of most IRES-containing RNAs as well as mRNAs coding ITAFs. HL-1 cardiomyocytes were transfected with gapmer *Neat1, Neat1_2,* or control. Polysomes were purified on sucrose gradient as described in Star Methods. The polysome profile is presented in ***Figure 8—figure supplement 1***. RNAs were purified from cytoplasmic extracts and from pooled polysomal fractions and analyzed

*Figure 8 continued on next page*

*Figure 8 continued*

on a Fluidigm deltagene PCR array from two biologicals replicates (cell culture dishes and cDNAs), each of them measured in three technical replicates (PCR reactions) (*Supplementary file 8*). IRES-containing mRNAs (**A–B**) and ITAF mRNA levels in polysomes (**C–D**) polysomal RNA/ total RNA were analyzed. Relative quantification (RQ) of mRNA level was calculated using the $2-_{\Delta\Delta CT}$ method with normalization to GAPDH mRNA and to HL-1 tranfected by gapmer control, and is shown as fold change of repression (red) or induction (blue).

The online version of this article includes the following figure supplement(s) for figure 8:

**Figure supplement 1.** Polysome profiles of HL-1 cardiomyocytes treated by gapmer *Neat1* or *Neat1_2*, compared to gapmer control.

**Figure supplement 2.** Effect of *Neat1* and *Neat1_2* knock-down on translation of mRNAs coding angiogenic and cardioprotective factors.

---

*Neat1* is not the first lncRNA to exhibit an ITAF function. The lncRNA TP53-regulated modulator of p27 (*TRMP*) has been recently described as an ITAF of the *Cdkn1b/p27^{kip}* IRES (*Yang et al., 2018*). TRMP inhibits the *p27^{kip}* IRES activity by competing with the IRES for pyrimidine tract binding protein (PTB) binding and prevents IRES activation mediated by PTB. Also, the lncRNA *ARAP-as1* directly interacts with SFPQ, which results in release of PTB and activation of *MYC* IRES (*Zhang et al., 2020*). We have not yet deciphered the mechanism of action of *Neat1*. We propose that the paraspeckle would be a recruitment platform for IRES-containing mRNAs. *Neat1*, by interacting with p54^{nrb} and other paraspeckle proteins/ITAFs, would thus allow IRESome formation in the paraspeckle (*Figure 9*). Is the role of *Neat1* exclusively nuclear in the paraspeckle, or is it exported to the cytoplasm with the IRESome complex? Several observations argue for the presence of *Neat1* in the cytoplasm: our FISH experiments clearly identify the *Neat1-2* isoform in the cytoplasm (*Figure 1—figure supplement 2*), while a recent report shows that *Neat1-1* isoform is released from nucleus to cytoplasm where it suppresses the Wnt signaling in leukemia stem cells and acts as a tumor suppressor in acute myeloid leukemia (*Yan et al., 2021*). *Neat1-2* isoform has been detected in the cytoplasm of hematopoietic cells by other authors. Interestingly, they identified a histone modifier, ASXL1, interacting with p54^{nrb}/ NONO and involved in paraspeckle formation. Mutation of ASXL1 generates *Neat1_2* export to the cytoplasm (*Yamamoto et al., 2021*). Furthermore, the role of cytoplasmic *Neat1* in translation is suggested by our previous data showing that *Neat1* is present in HL-1 cell polysomes and that this association with polysomes is increased in early hypoxia (*Hantelys et al., 2019*). The involvement of *Neat1* in translation control via a cytoplasmic location is also supported by the presence of the triple helix in the 3'UTR of *Neat1_2*, whose role in translation activation has been demonstrated (*Wilusz et al., 2012*).

The model of IRESome formation mediated by *Neat1* in the paraspeckle, and the absence of any impact of *Neat1* on the picornaviral EMCV IRES activity, are both consistent with previous reports suggesting that the site of mRNA synthesis is crucial for IRES structure and function (*Semler and Waterman, 2008*). For picornaviruses whose mRNAs are synthesized in the cytoplasm, IRES elements would be able to form an IRESome RNP in the cytoplasm. In contrast, cellular mRNAs (as well as DNA viruses and retroviruses mRNAs) transcribed in the nucleus need a nuclear event (*Ainaoui et al., 2015; Stoneley et al., 2000*). The present data provide a mechanism for this nuclear history and reveal a new function of the paraspeckle, a nuclear body, in IRESome formation (*Figure 9*).

A role of *Neat1* in ischemic heart has been recently reported showing that *Neat1* down-regulation would protect cardiomyocytes from apoptosis by regulating the processing of pri-miR-22 (*Gidlöf et al., 2020*). Surprisingly, these authors show that hypoxia down-regulates *Neat1* expression in cardiomyocytes. This contradicts our data showing that *Neat1* is induced by hypoxia. Our data are however in agreement with the rest of the literature showing that Neat1 is induced by hypoxia in tumors, its transcription being activated by HIF-2 (*Choudhry et al., 2015*). Another study also showed that *Neat1* overexpression protects cardiomyocytes against apoptosis by sponging miR125a-5p, resulting in upregulation of the apoptosis repressor gene B-cell lymphoma-2-like 12 (BCL2L12) (*Yan et al., 2019*). These contradictory reports highlight the complex impact of *Neat1* on miRNA-mediated gene regulation.

In the present study, we have uncovered a novel role of *Neat1* in the translational control of several families of genes involved in stress response, angiogenesis and cardioprotection, while it does not affect global translation. The increased protein synthesis from mRNAs coding ITAFs favors a wide

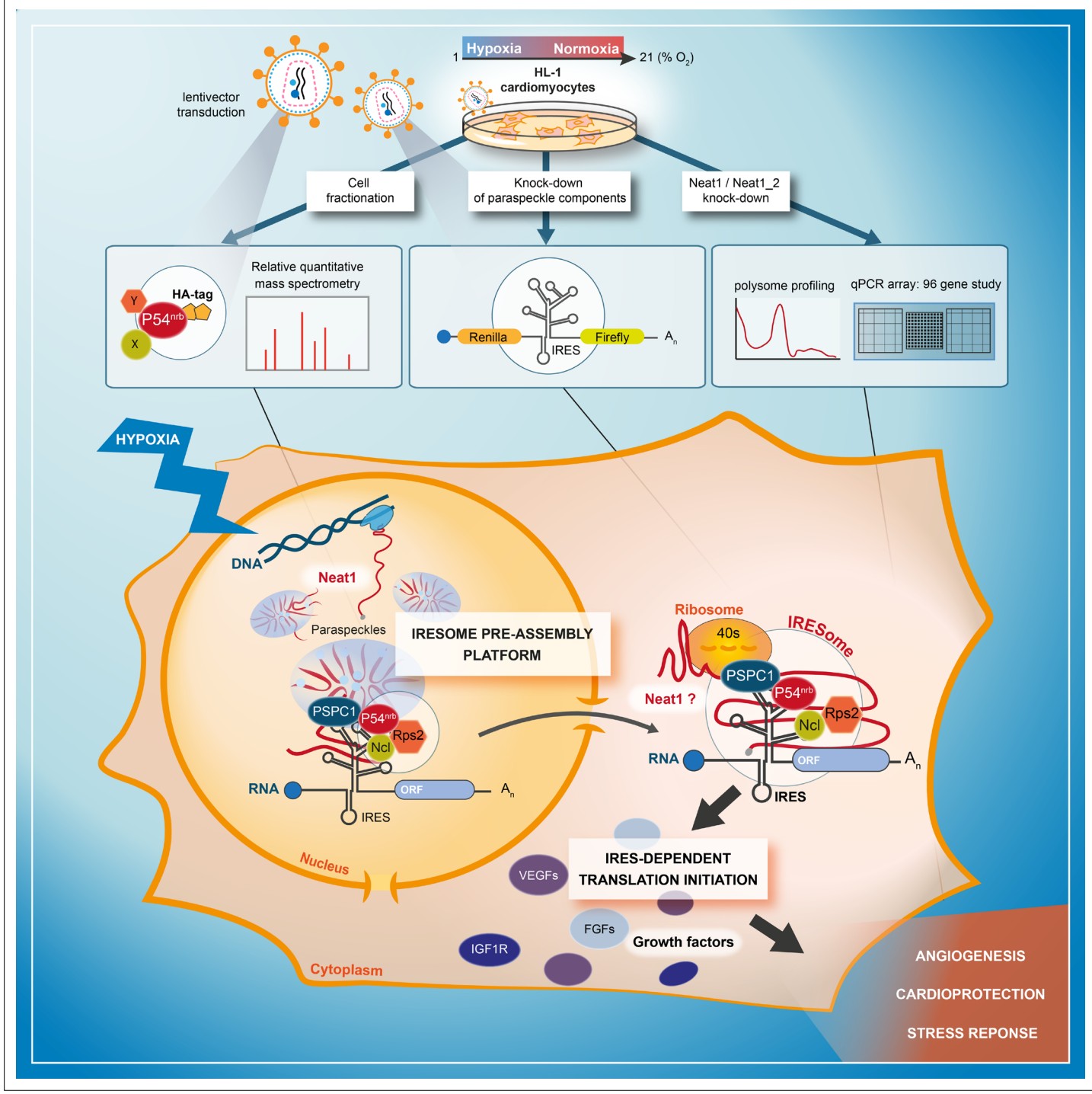

**Figure 9.** Model of IRESome formation in the paraspeckle. According to the present data, we propose that the paraspeckle may be a recruitment platform for IRES-containing mRNAs in hypoxic cardiomyocytes. *Neat1* and proteins present in the paraspeckle (among them major paraspeckle components such as p54nrb and PSPC1) would assemble the IRESome, then mRNA would be exported from the nucleus and translated in the cytosol. Identification of Neat1 in the cytoplasm suggests that it might be part of the IRESome and have a direct role in translation. However this latter hypothesis remains to be elucidated.

role of *Neat1* and of the paraspeckle in activating IRES-dependent translation. Many of the genes involved in angiogenesis or cardioprotection tested here have not been described as containing an IRES in their mRNAs. We can make the hypothesis that these mRNA families either contain IRESs that have not been identified yet, or are translated by another cap-independent mechanism such as m6A-induced ribosome engagement sites (MIRES) (*Prats et al., 2020*).

*Neat1*, as a stress-induced lncRNA, plays a role in many pathologies including cancer and ischemic diseases, thus its central role in the translational control of expression of genes involved in tissue revascularization and cell survival makes it a potential therapeutic target of great interest.

# Materials and methods

**Key resources table**

| Reagent type (species) or resource | Designation | Source or reference | Identifiers | Additional information |
|---|---|---|---|---|
| Antibody | Anti-P54nrb (rabbit polyclonal) | Santacruz | Sc-67016 | Dilution 1:200 (capillary Western) Dilution 1:400 (classical Western) |
| Antibody | Anti-PSPC1 (rabbit polyclonal) | bethyl laboratory | A303-205A | Dilution 1:100 (capillary Western) Dilution 1:1000 (classical Western) |
| Antibody | Anti-SFPQ (mouse monoclonal) | Abcam | Ab11825 | Dilution 1:100 |
| Antibody | Anti-FGF1 (rabbit polyclonal) | Abcam | Ab207321 | Dilution 1:25 |
| Antibody | Anti-nucleolin (rabbit polyclonal) | Novus biological | NB600-241 | Dilution 1:50 |
| Antibody | Anti-Histone H3 (rabbit polyclonal) | Cell Signaling | 4499 | Dilution 1 : 10000 |
| Antibody | Anti-GAPDH (mouse monoclonal) | SantaCruz | Sc-32233 | Dilution 1:1000 |
| Antibody | Mouse total IgG (mouse polyclonal) | Sigma | I5381 | 2 mg/mL |
| Antibody | Anti-eIF2α (rabbit polyclonal) | Cell Signaling Technology | 9721 | Dilution 1:50 |
| Antibody | Anti-phospho-eIF2α (mouse monoclonal) | Cell Signaling Technology | 2103 | Dilution 1:50 |
| Antibody | Anti-p21 (mouse monoclonal) | Santacruz | Sc-6246 | Dilution 1:50 |
| Antibody | Anti-HA (mouse monoclonal) | Sigma | H9558/H3663 | 2.4 mg/mL (72 µg) |
| Antibody | Anti-rabbit-peroxidase conjugate (donkey polyclonal) | Jackson ImmunoResearch | 711-035-152 | Dilution 1:10000 |
| Antibody | Anti-mouse-peroxidase conjugate (rabbit polyclonal) | Jackson ImmunoResearch | 715-035-150 | Dilution 1:10000 |
| Antibody | Rabbit detection module | Protein Simple | DM-001 | 10 µl |
| Antibody | Mouse detection module | Protein Simple | DM-002 | 10 µl |
| Strain, strain background (*Escherichia coli*) | Top10 | InVitrogen | C404003 | |

*Continued on next page*

Continued

| Reagent type (species) or resource | Designation | Source or reference | Identifiers | Additional information |
|---|---|---|---|---|
| Strain, strain background (*Escherichia coli*) | Strataclone | Agilent technologies | 200185 | |
| Chemical compound, drug | TRI-Reagent | MRC Inc | TR118 | |
| Chemical compound, drug | Isopropanol | Sigma-Aldrich | 33539 | |
| Chemical compound, drug | Ethanol | Sigma-Aldrich | 32221 | |
| Chemical compound, drug | Digitonin | Sigma-Aldrich | D141 | |
| Chemical compound, drug | NP40 (IGEPAL 630) | Sigma-Aldrich | I8896 | |
| Chemical compound, drug | EDTA | Euromedex | EU0084-A | |
| Chemical compound, drug | Proteinase inhibitor cocktail | Sigma-Aldrich | P2714 | |
| Chemical compound, drug | RNAse inhibitor | AppliedBiosystem | N8080119 | |
| Chemical compound, drug | Formamide | Invitrogen | 15515026 | |
| Chemical compound, drug | Paraformaldehyde 16% | Electron Microscopy Science | | |
| Chemical compound, drug | SSC saline-sodium citrate buffer | Euromedex | EU0300-C | |
| Chemical compound, drug | RIPA | BioBasic | RB4476 | |
| Peptide, recombinant protein | HA peptides | Sigma-Aldrich | I2149 | |
| Commercial assay or kit | Premix Ex Taq II | Takara | RR820B | |
| Commercial assay or kit | EZ view red protein G beads | Sigma | E3403 | |
| Commercial assay or kit | DG32 cartridge | Bio-Rad | #1864108 | |
| Commercial assay or kit | QX200 ddPCR EvaGreen Supermix | Bio-Rad | 1864034 | |
| Commercial assay or kit | High capacity cDNA Reverse transcription kit | Thermofisher | 4368814 | |
| Commercial assay or kit | NucleoBond Xtra Maxi kits | Macherey-Nagel | 740414.10 | |
| Commercial assay or kit | EZ-10 Spin Column Plasmid DNA Miniprep Kit | BioBasic | BS413 | |
| Commercial assay or kit | StrataClone Blunt PCR Cloning Kit | Agilent | 240207 | |
| Commercial assay or kit | Dual-Luciferase Reporter Assay system | Promega | E1980 | |
| Commercial assay or kit | Jess or Wes Separation Module | ProteinSimple | SM-SW004 | |

*Continued on next page*

*Continued*

| Reagent type (species) or resource | Designation | Source or reference | Identifiers | Additional information |
|---|---|---|---|---|
| Commercial assay or kit | Fluorescent 5 x Master Mix 1 | ProteinSimple | PS-FL01-8 | |
| Cell line (*Homo-sapiens*) | 293 FT | Invitrogen | R700-07 | |
| Cell line (*Homo-sapiens*) | HT1080 | ATCC | CCL-121 | |
| Cell line (*Mus musculus*) | HL-1 | (*Claycomb et al., 1998*) / Sigma-Aldrich | SCC065 | Beating cardiomyocytes (*Video 1*) |
| Sequence-based reagent | NEAT1 | This paper | PCR primers | *Supplementary file 2* |
| Sequence-based reagent | FGF1 | This paper | PCR primers | *Supplementary file 2* |
| Sequence-based reagent | NEAT1_2 | This paper | PCR primers | *Supplementary file 2* |
| Sequence-based reagent | HPRT | This paper | PCR primers | *Supplementary file 2* |
| Sequence-based reagent | RPL11 | This paper | PCR primers | *Supplementary file 2* |
| Sequence-based reagent | 18 S | *Hantelys et al., 2019* | PCR primers | *Supplementary file 2* |
| Sequence-based reagent | SFPQ | This paper | PCR primers | *Supplementary file 2* |
| Sequence-based reagent | P54nrb | This paper | PCR primers | *Supplementary file 2* |
| Sequence-based reagent | PSPC1 | This paper | PCR primers | *Supplementary file 2* |
| Sequence-based reagent | NUCLEOLIN | This paper | PCR primers | *Supplementary file 2* |
| Sequence-based reagent | RPS2 | This paper | PCR primers | *Supplementary file 2* |
| Sequence-based reagent | HNRNPM | *Hantelys et al., 2019* | PCR primers | *Supplementary file 2* |
| Sequence-based reagent | Fluidigm deltagene probes | This paper | PCR primers | *Supplementary file 2* |
| Sequence-based reagent | *Neat1* and *Neat1_2* FISH probes | This paper | Hybridization probes | *Supplementary file 2* |
| Sequence-based reagent | SmiFISH secondary probes (FLAP X-Cy3 and FLAP-Y-Cy5) | This paper | Hybridization probes | *Supplementary file 2* |
| Sequence-based reagent | SmiFISH Neat1 primary probes | This paper | Hybridization probes | *Supplementary file 2* |
| Sequence-based reagent | SmiFISH bicistronic Lucky Luke mRNA primary probes | This paper | Hybridization probes | *Supplementary file 2* |
| Sequence-based reagent | HA-p54[nrb] | This paper | Cloning primers | *Supplementary file 2* |
| Sequence-based reagent | miR-Neat1-G2 | This paper | Cloning primers | *Supplementary file 2* |

*Continued*

| Reagent type (species) or resource | Designation | Source or reference | Identifiers | Additional information |
|---|---|---|---|---|
| Sequence-based reagent | miR-Neat1_2-G6 | This paper | Cloning primers | *Supplementary file 2* |
| Sequence-based reagent | miR-Neat1_2-G7 | This paper | Cloning primers | *Supplementary file 2* |
| Sequence-based reagent | P54nrb mouse | Dharmacon E-048587-01-0005 | siRNA smartpool | *Supplementary file 2* |
| Sequence-based reagent | PSPC1 mouse | Dharmacon E-049216-00-0005 | siRNA smartpool | *Supplementary file 2* |
| Sequence-based reagent | SFPQ mouse | Dharmacon E-044760-00-0005 | siRNA smartpool | *Supplementary file 2* |
| Sequence-based reagent | Nucleolin mouse | Dharmacon E-059054-00-0005 | siRNA smartpool | *Supplementary file 2* |
| Sequence-based reagent | Rps2 mouse | Dharmacon E-049205-00-0005 | siRNA smartpool | *Supplementary file 2* |
| Sequence-based reagent | hnRNPM mouse | Dharmacon E-044465-00-0005 | siRNA smartpool | *Supplementary file 2* |
| Sequence-based reagent | siRNA non-targeting control | Dharmacon D-001910-10-20 | siRNA | *Supplementary file 2* |
| Sequence-based reagent | NEAT1 A | LG00218175 | LNA gapmer | *Supplementary file 2* |
| Sequence-based reagent | NEAT1 B | LG00218176 | LNA gapmer | *Supplementary file 2* |
| Sequence-based reagent | NEAT1 C | LG00218177 | LNA gapmer | *Supplementary file 2* |
| Sequence-based reagent | NEAT1 D | LG00218178 | LNA gapmer | *Supplementary file 2* |
| Sequence-based reagent | NEAT1_2 | LG00234548 | LNA gapmer | *Supplementary file 2* |
| Sequence-based reagent | NEGATIVE CONTROL | LG00000002 | LNA gapmer | *Supplementary file 2* |
| Recombinant DNA reagent | pTRIP-CRHL+ | Sequence available on Dryad, (2) | SIN lentivector plasmid | doi:10.5061/dryad.nvx0k6dq7 |
| Recombinant DNA reagent | pTRIP-CRF1AL+ | Sequence available on Dryad, (17; 26) | SIN lentivector plasmid | doi:10.5061/dryad.nvx0k6dq7 |
| Recombinant DNA reagent | pTRIP-CRFL+ | Sequence available on Dryad, (25) | SIN lentivector plasmid | doi:10.5061/dryad.nvx0k6dq7 |
| Recombinant DNA reagent | pTRIP-CRVAaL+ | Sequence available on Dryad, (16) | SIN lentivector plasmid | doi:10.5061/dryad.nvx0k6dq7 |
| Recombinant DNA reagent | pTRIP-CRVAbL+ | Sequence available on Dryad, (16) | SIN lentivector plasmid | doi:10.5061/dryad.nvx0k6dq7 |
| Recombinant DNA reagent | pTRIP-CRhVCL+ | Sequence available on Dryad, (2) | SIN lentivector plasmid | doi:10.5061/dryad.nvx0k6dq7 |
| Recombinant DNA reagent | pTRIP-CRhVDL+ | Sequence available on Dryad, (13) | SIN lentivector plasmid | doi:10.5061/dryad.nvx0k6dq7 |
| Recombinant DNA reagent | pTRIP-CRMP2L+ | Sequence available on Dryad, (42) | SIN lentivector plasmid | doi:10.5061/dryad.nvx0k6dq7 |
| Recombinant DNA reagent | pTRIP-CREL+ | Sequence available on Dryad, (25) | SIN lentivector plasmid | doi:10.5061/dryad.nvx0k6dq7 |

*Continued*

| Reagent type (species) or resource | Designation | Source or reference | Identifiers | Additional information |
|---|---|---|---|---|
| Recombinant DNA reagent | pTRIP-CRIGL+ | This paper | SIN lentivector plasmid | doi:10.5061/dryad.m0cfxpp75 |
| Recombinant DNA reagent | pCMV-dR8.91 | Addgene | Plasmid for lentivector production | |
| Recombinant DNA reagent | pCMV-VSV-G | Addgene | Plasmid for lentivector production | |
| Recombinant DNA reagent | pTRIP-Neat1-miR-G2 | This paper | SIN lentivector plasmid | doi:10.5061/dryad.m0cfxpp75 |
| Recombinant DNA reagent | pTRIP-Neat1_2-miR-G6 | This paper | SIN lentivector plasmid | doi:10.5061/dryad.m0cfxpp75 |
| Recombinant DNA reagent | pTRIP-Neat1_2-miR-G7 | This paper | SIN lentivector plasmid | doi:10.5061/dryad.m0cfxpp75 |
| Recombinant DNA reagent | pTRIP-HA2-P54nrb | This paper | SIN lentivector plasmid | doi:10.5061/dryad.m0cfxpp75 |
| Software, algorithm | Prism 6 | Graphpad | Software to perform statistics | https://www.graphpad.com/scientific-software/prism/ |
| Software, algorithm | Excel 2007 | Microsoft office | Software to perfom graphs and tables | |
| Software, algorithm | FIJI | FIJI | Software for image analysis | https://fiji.sc/ |
| Software, algorithm | ImageJ | ImageJ/NIH | Software for image analysis | https://imagej.nih.gov/ij/download.html |
| Software, algorithm | Zen black/Blue edition | Zeiss | Microscope software | https://www.zeiss.fr/microscopie/produits/microscope-software/zen-lite.html |
| Software, algorithm | QuantStudio | AppliedBiosystems | Quantification software | https://www.thermofisher.com/fr/fr/home/global/forms/life-science/quantstudio-3-5-software.html |
| Software, algorithm | QuantaSoft 1.7.4 | Bio-Rad | Western blot quantification software | https://www.bio-rad.com/fr-fr/sku/1864011-quantasoft-software-regulatory-edition?ID=1864011 |
| Software, algorithm | Microwin 2000 | Berthold | Microplaque testing software | https://fr.freedownloadmanager.org/Windows-PC/MikroWin-2000.html |
| Software, algorithm | LSM780 Zeiss confocal microscope | Zeiss | Microscope software | N/A |
| Software, algorithm | Compass for SW | Protein Simple | Capillary Western software | N/A |

## Lead contact and materials availability

Further information and requests for resources and reagents should be directed to and will be fulfilled by the Lead Contact, Anne-Catherine Prats (anne-catherine.prats@inserm.fr).

## Experimental model and subject details
### Cell lines

Female human embryonic kidney cells HEK-293FT (Invitrogen R700-07) and male human fibrosarcoma HT1080 cells (ATCC CCL-121) were cultured in DMEM-GlutaMAX +Pyruvate (Life Technologies SAS, Saint-Aubin, France), supplemented with 10% fetal bovine serum (FBS), and MEM essential and non-essential amino acids (Sigma-Aldrich). They were characterized by the supplier, then by their capacity to be transfected efficiently to produce and titrate lentivectors. Female mouse atrial HL-1 cardiomy-ocytes (Sigma-Aldrich SCC065) were extensively characterized by the supplier and by ourselves, by their beating phenotype (*Video 1*). They were cultured in Claycomb medium containing 10% FBS, Penicillin/Streptomycin (100 U/mL-100µg/mL), 0.1 mM norepinephrine, and 2 mM L-Glutamine. Cell culture flasks were precoated with a solution of 0.5% fibronectin and 0.02% gelatin for 1 hr overnight

at 37 °C (Sigma-Aldrich). To keep HL-1 phenotype, cell culture was maintained as previously described (*Claycomb et al., 1998*). All cells were cultured in a humidified chamber at 37 °C and 5% $CO_2$. When subjected to hypoxia, cells were incubated at 37 °C under 1% $O_2$. All cell types were tested negative for mycoplasma contamination every three months with the MycoAlert Mycoplasma Detection Kit (Lonza).

## Bacterial strains

- Top 10 *Escherichia coli* (InVitrogen, thermofisher scientific C404003)
- Strataclone *Escherichia coli* (Agilent technologies, 200185)

These cells were stored at –80 °C and grown in LB medium at 37 °C. Top10 cells were used for plasmid amplification of pTRIP lentivector. Strataclone cells were used for recombination and amplification of PCR product into pSC-B-amp/kan plasmid.

## Method details

### Cell transfection

siRNA treatment on transduced cells was performed 72 hr after transduction (and after one cell passage) in 24-well plates for reporter activity assay or 12 well plates for gene expression experiments. HL-1 were transfected by siRNAs as follows: one day after being plated, cells were transfected with 10 nM of small interference RNAs from Dharmacon Acell SMARTpool targeting P54[nrb], PSPC1, SFPQ, hnRNPM, Nucleolin, RPS2, or non-targeting siRNA control (siControl), using INTERFERin (Polyplus Transfection) according to the manufacturer's recommendations, in DMEM-GlutaMAX +Pyruvate media without penicillin-streptomycin. The media was changed 24 hr after transfection and the cells were incubated 72 hr for the time of transfection at 37 °C with siRNA. For *Neat1* knock-down, HL-1 cells were transduced with a pool of 4 gapmers (Qiagen) at 40 nM (10 nM each) and incubated 48 hr after transfection, proceeded essentially as described above (siRNA and gapmer sequences are provided in *Supplementary file 2*).

### Cell transduction

For lentivector transduction, HL-1 cardiomyocytes were plated into a T25 flask and transduced overnight in 2.5 mL of transduction medium (OptiMEM-GlutaMAX, Life Technologies SAS) containing 5 μg/mL protamine sulfate in the presence of lentivectors (MOI 2). HL-1 cells were transduced with an 80–90% efficiency in the mean.

### Lentivector construction

Bicistronic lentivectors coding for the renilla luciferase (LucR) and the stabilized firefly luciferase Luc+ (called LucF in the text) were constructed from the dual luciferase lentivectors described previously, which contained Luc2CP (*Morfoisse et al., 2014*; *Morfoisse et al., 2016*). The LucR gene used here is a modified version of LucR where all the predicted splice donor sites have been mutated. The cDNA sequences of the human *FGF1, –2, VEGFA, -C, -D, MYC* and EMCV IRESs were introduced between the first (LucR) and the second cistron (LucF) (*Giraud et al., 2001*; *Nanbru et al., 1997*; *Prats et al., 2013*; *Vagner et al., 1995*). IRES sequence sizes are: 430 nt (*FGF1*), 480 nt (*FGF2*), 302 nt (*VEGFAa*), 485 nt (*VEGFAb*), 419 nt (*VEGFC*), 507 nt (*VEGFD*), 363 nt (*c-MYC*), 640 nt (EMCV), 973 nt (rat *IGF1R*) (*Huez et al., 1998*; *Martineau et al., 2004*; *Morfoisse et al., 2014*; *Morfoisse et al., 2016*; *Nanbru et al., 1997*; *Vagner et al., 1995*). The two IRESs of the *VEGFA* mRNA have been used and are called *VEGFA*a and *VEGFA*b, respectively (*Huez et al., 1998*). The hairpin negative control contains a 63 nt long palindromic sequence cloned between LucR and LucF genes (*Hantelys et al., 2019*). This control has been successfully validated in previous studies (*Créancier et al., 2000*; *Morfoisse et al., 2014*). The expression cassettes were inserted into the SIN lentivector pTRIP-DU3-CMV-MCS vector described previously (*Prats et al., 2013*). All cassettes are under the control of the cytomegalovirus (CMV) promoter. The lentivectors coding artificial miRNAs miR-Neat1 and miR-*Neat1_2* were constructed by inserting double-stranded oligonucleotides targeting *Neat1* or *Neat1_2*, according to a protocol adapted from the BLOCK-iT technology of Life Technologies (sequences provided in *Supplementary file 2*). The lentivector coding HA-p54[nrb] was obtained by amplifying the p54 cDNA by PCR with a forward primer containing the sequence of two HA motifs. The resulting fragment was

cloned into the pTRIP vector. Plasmid construction and amplification were performed in the bacteria strain TOP10 (Thermofisher Scientific, Illkirch Graffenstaden, France). Vector sequences are available on Dryad (doi:10.5061/dryad.nvx0k6dq7 or doi:10.5061/dryad.m0cfxpp75).

## Lentivector production

Lentivector particles were produced using the $CaCl_2$ method based by tri-transfection with the plasmids pCMV-dR8.91 and pCMV-VSVG, $CaCl_2$ and Hepes Buffered Saline (Sigma-Aldrich, Saint-Quentin-Fallavier, France), into HEK-293FT cells. Viral supernatants were harvested 48 hr after transfection, passed through 0.45 μm PVDF filters (Dominique Dutscher SAS, Brumath, France), and stored in aliquots at –80 °C until use. Viral production titers were assessed on HT1080 cells with serial dilutions of a lentivector expressing GFP and scored for green fluorescent protein (GFP) expression by flow cytometry analysis on a BD FACSVerse (BD Biosciences, Le Pont de Claix, France).

## Reporter activity assay

For reporter lentivectors, luciferase activities were performed in vitro and in vivo were performed using Dual-Luciferase Reporter Assay (Promega, Charbonnières-Les-Bains, France). Briefly, proteins from HL-1 cells were extracted with Passive Lysis Buffer (Promega France). Bioluminescence was quantified with a luminometer (Centro LB960, Berthold, Thoiry, France) from 9 to 12 biological replicates and with three technical replicates for each sample in the analysis plate.

## FISH

HL-1 cells were cultured in 12-well plates on fibronectin-gelatin coated 15 mm coverglass 1.5 thickness (Menzel-Gläser). FISH probes were produced and purchased from Sigma-Aldrich, and delivered HPLC purified at 50 nmol. The 3/2 probes used per target (*Neat1* and *Neat1_2* isoform respectively) are between 38 and 40 mer long and are conjugated to one Cy3 through 5' amino acid modifications (see *Supplementary file 2* for sequences).

FISH was performed as previously described (http://www.singerlab.org/protocols). Briefly, cells were fixed with 4% paraformaldehyde (electron microscopy science), rinsed twice, and permeabilized overnight in 70% ETOH. Then cells were pre-hybridized in a 15% formamide/2 X SSC buffer at room temperature. The hybridization reaction was performed overnight at 37 °C with a Mix of 2XSSC, 0.5 mg/mL yeast tRNA, 15% formamide, 10% dextran sulfate, and 10 ng of mixed probes. Then the coverslip was rinsed two times 10 min in 2 X SSC and 1XSSC for 10 min, before mounting on Moviol mounting medium supplemented with DAPI. Three-dimensional image stacks were captured on LSM780 Zeiss confocal microscope, camera lens x63 with Z acquisition of 0.45 μM, and Zen software (Zeiss).

## SmiFISH

A set of target-specific primary probes was produced and purchased from Integrated DNA Technologies (IDT). Each probe carried an additional 28 nt-long sequence 'FLAP' which is not represented in either mouse or human genomes. The primary probes against bicistronic mRNA were complementary to the fluorescent secondary probe FLAP-X, and the primary probes Neat1 were complementary to the fluorescent secondary probe FLAP-Y. The two secondary probes FLAP-X and FLAP-Y were also from IDT, conjugated to fluorophores Cy3 and Cy5, respectively. All probe sequences are presented in *Supplementary file 2*. SmiFISH was performed as previously described (*Tsanov et al., 2016*). Cells were grown to 80% confluence in six-well plates and subjected or not to 4 hr of hypoxia. Cells were fixed with 4% paraformaldehyde for 20 min (electron microscopy science) at room temperature, rinsed twice and permeabilized overnight in 70% ETOH. Cy3 and Cy5-labeled fluorescent FLAPs were pre-annealed to primary probes prior to in situ hybridization. Then cells were pre-hybridized with probes (40 pmol) in a 15% formamide/1 X SSC buffer at 37 °C. The hybridization reaction was performed overnight at 37 °C with Mix 1 (2XSSC, 25 μg/μL yeast tRNA, 100% formamide, FLAP-structured duplex (FLAP-Y duplex +FLAP X duplex) and H2O)+Mix 2 (20 mg/mL RNAse free BSA, 200 mM VRC, 40% dextran sulfate and H2O). Then the coverslip was rinsed five times in 1 X SSC 15% formamide mix at 37 °C and twice in PBS before mounting on Dako mounting medium supplemented with DAPI. Three-dimensional image stacks were captured on Zeiss Axiomager Z3 Apotome confocal microscope,

camera lens x63 with Z acquisition of 0.45 µM, and Zen software (Zeiss). For *Figure 2*, images were analyzed with a script for ImageJ. For each segmented nucleus, spots were segmented by detecting local maxima after applying a laplacien filter. For *Figure 3*, images were analyzed with IMARIS. For each image, spots were detected using the 'spot' function and the colocalization with the 'co-localize spots' function.

## Western blot

Cells were harvested on ice, washed with cold PBS, and collected on RIPA buffer Biobasic supplemented with protease inhibitor (Sigma). Protein concentration was measured using BCA Protein Assay Kit (Interchim), and equal amounts of proteins were subjected to SDS-PAGE (TGX Stin Free Fast-Cast Acrylamid, 12%, Bio-Rad, 161–0185) and transferred onto nitrocellulose membrane (Transblot Turbo, Bio-Rad, 1704271). Membranes were washed in Tris-buffered saline supplemented with 0.05% Tween-20 and then saturated in Tris-buffered saline supplemented with 0.05% Tween-20 with 5% BSA, incubated overnight with primary antibodies in Tris-buffered saline supplemented with 0.05% Tween-20 with 5% BSA, washed and revealed with Clarity Western ECL Substrate (Bio-Rad, 170–5060). Western blotting was conducted using standard methods with the following antibodies: Rabbit anti-PSPC1 (bethyl laboratory, A303-205A) diluted 1:1000, Rabbit anti-P54[nrb] (Santacruz, sc67016) diluted 1/400, Rabbit Histone H3 (Cell Signaling, 4499) diluted 1/10000, mouse GAPDH (SantaCruz, SC32233) diluted 1/1000, secondary donkey anti-rabbit IgG antibody, Peroxidase Conjugated, (Jackson ImmunoResearch, 711-035-152) diluted 1:10000, secondary rabbit anti-mouse IgG antibody, Peroxidase Conjugated, (Jackson ImmunoResearch, 715-035-150) diluted 1:10000.

## Capillary Western

Diluted protein lysate was mixed with fluorescent master mix and heated at 95 °C for 5 min. Three µL of protein mix (1 mg/mL maximal concentration) containing Protein Normalization Reagent, blocking reagent, wash buffer, target primary antibody (rabbit anti-eIF2α [Cell Signaling Technology 9721]) diluted 1:50, mouse anti-phospho-eIF2α [Cell Signaling Technology 2103] diluted 1:50, mouse anti-p21 antibody [Santacruz, sc-6246] diluted 1:50, rabbit anti-P54[nrb] diluted 1:200 [Santacruz, sc-67016], rabbit anti-PSPC1 diluted 1:100 [bethyl laboratory, A303-205A], mouse anti-SFPQ diluted 1:100 [Abcam, Ab11825]; rabbit anti-FGF1 diluted 1:25 [Abcam Ab207321], rabbit anti-Nucleolin diluted 1:50 [Novus biological, NB600-241], secondary-HRP (ready to use rabbit or mouse 'detection module', DM-001 or αDM-002), and chemiluminescent substrate were dispensed into designated wells in a manufacturer-provided microplate. The plate was loaded into the instrument (Jess, Protein Simple) and proteins were drawn into individual capillaries on a 25 capillary cassette (12–230 kDa) (SM-SW004). Normalization reagent allow detecting total protein in the capillary through the binding of amine group by a biomolecule and to get rid of housekeeping protein that can arbor an inconsistent and unreliable expression. Graph plotted in Figures represent chemiluminescence value before normalization.

## Measurement of protein half-life

To measure protein half-life, HL-1 cardiomyocytes were treated with cycloheximide (InSolution CalBioChem) diluted in PBS to a final concentration of 10 µg/mL in six-well plates. Time-course points were taken by stopping cell cultures after 0 hr, 30 min, 1 hr, 2 hr, 4 hr, 8 hr, 16 hr, or 24 hr of incubation and subsequent capillary Western analysis of cell extracts.

## RNA purification and cDNA synthesis

Total RNA extraction from HL-1 cells was performed using TRI Reagent according to the manufacturer's instructions (Molecular Research Center Inc, USA). RNA quality and quantification were assessed by a Nanodrop spectrophotometer (Nanodrop 2000, Thermo Scientific). 750 ng RNA was used to synthesize cDNA using a High-Capacity cDNA Reverse Transcription Kit (Applied Biosystems, France). Appropriate no-reverse transcription and no-template controls were included in the qPCR assay plate to monitor potential reagent or genomic DNA contaminations, respectively. The resulting cDNA was diluted 10 times in nuclease-free water. All reactions for the PCR array were run in biological triplicates.

## qPCR

7.5 ng cDNA were mixed with 2X TB green Premix Ex Taq II (Takara, RR820B), 10 µM forward and reverse primers, according to manufacturer instruction. qPCR reactions were performed on Viia7 (Applied Biosystems) and the oligonucleotide primers used are detailed in *Supplementary file 2*. The reference genes were *Hprt*, *18* S and/or *Rpl11*.

## ddPCR

ddPCR reaction for *Neat1* knock-down control were performed with the Bio-Rad system. The ddPCR reaction mixture (22 µl) contained 2 x QX200 ddPCR EvaGreen Supermix (no dUTP) (Bio-Rad), 2 µM of a mix of forward and reverse primers (*Supplementary file 2*), and 2/4/6 µL of cDNA depending on the target. The reaction mixture was transferred for droplet generation by AutoDG System (Bio-Rad) in individual wells of disposable DG32 Automated Droplet Generator Cartridges that were already placed in the cartridge holder. The droplet was generated by AutoDG System, between 15000–20000 droplets/well. The prepared droplet emulsions were further loaded in ddPCR 96-Well Plates (Bio-rad) by aspirating 40 µl from the DG32 cartridge by the AutoDG System. The plate was then heat sealed with pierceable foil using a PX1 PCR plate sealer 5 s at 180 °C (Bio-Rad), and PCR amplification was carried out in a T100 thermal cycler (Bio-Rad). The thermal consisted of initial denaturation at 95 °C for 5 min followed by 40 cycles of 95 °C for 30 s (denaturation) and 60 °C for 1 minute (annealing/ elongation) with a ramp of 2 °C/s, a signal stabilization step at 4 °C 5 min followed by 90 °C 5 min. After PCR amplification the positive droplets were counted with a QX200 droplet reader (Bio-Rad).

## Cell fractionation

HL-1 cells placed in normoxia or hypoxia and transduced by P54nrb-HA construct were trypsinized, rinsed with PBS and lysed in solution 1 (Hepes 50 mM/NaCl 150 mM pH7.3, digitonin (100 µg/mL), EDTA 1 mM, protease inhibitor cocktail) and incubated on ice. Then the lysate was centrifugated at 2000 g for 5 min and the supernatant (cytosolic fraction) was aliquoted. Then the pellet was rinsed in PBS, and incubated in solution 2 (Hepes 50 mM/NaCl 150 mM pH7.3, NP40 1%, EDTA 1 mM, protease inhibitor cocktail) during 30 min at 4 °C. After centrifugation at 7000 g, the pellet was rinsed and resuspended in solution 3 (Tris/HCl 50 mM, NaCl 150 mM, NP40 1%, sodium deoxycholate 0.5%, SDS 0.1% (RIPA), protease inhibitor cocktail) and incubated for 10 min at 4 °C. Finally, the lysate was centrifuged for 10 min at 8200 g and the supernatant was aliquoted (nuclear fraction).

## Immunoprecipitation

Immunoprecipitation experiments were realized with 150 µg of total protein amounts from the cytosolic and nuclear fraction in normoxia or hypoxia, with a HA antibody (H9558/H3663, Sigma) 72 µg (2.4 mg/mL) or IgG mouse control (Sigma I5381) (2 mg/mL) using EZ view red protein G beads (Sigma). The beads-antibody-protein mix was incubated overnight at 4 °C and bounds protein were eluted with 35 µg HA peptides diluted in PBS (Sigma); then Laemmli buffer was added and the eluate heated at 95 °C 2 min.

## In-gel trypsin digestion and mass spectrometry analysis

For mass spectrometry analysis, immunoprecipitated samples, prepared in triple or quadruple biological replicates for each condition, were submitted to an additional protein reduction in 24.5 mM dithiothreitol for 30 min at 56 °C followed by alkylation of cysteine residues in 74 mM iodoacetamide for 30 min in the dark at room temperature. Each reduced/alkylated sample was loaded onto 1D SDS-PAGE gel (stacking 4% and separating 12% acrylamide). For a one-shot analysis of the entire mixture, no fractionation was performed, and the electrophoretic migration was stopped as soon as the protein sample entered the separating gel. The gel was briefly stained using Quick Coomassie Blue (Generon). Each single slice containing the whole sample was excised and subjected to in-gel tryptic digestion using modified porcine trypsin (Promega, France) at 10 ng/µl as previously described (*Shevchenko et al., 1996*). The dried peptide extracts obtained were dissolved in 17 µl of 0.05% trifluoroacetic acid in 2% acetonitrile and analyzed by online nanoLC using an Ultimate 3000 RSLCnano LC system (Thermo Scientific Dionex) coupled to an LTQ Orbitrap Velos mass spectrometer (Thermo Scientific, Bremen, Germany) for data-dependent CID fragmentation experiments. Five µl of each

peptide extracts were loaded in two or three injection replicates onto 300 µm ID x 5 mm PepMap C18 pre-column (ThermoFisher, Dionex) at 20 µl/min in 2% acetonitrile, 0.05% trifluoroacetic acid. After 5 min of desalting, peptides were online separated on a 75 µm ID x 50 cm C18 column (in-house packed with Reprosil C18-AQ Pur 3 µm resin, Dr. Maisch; Proxeon Biosystems, Odense, Denmark), equilibrated in 95% of buffer A (0.2% formic acid), with a gradient of 5 to 25% of buffer B (80% aceto-nitrile, 0.2% formic acid) for 80 min then 25% to 50% for 30 min at a flow rate of 300 nL/min. The LTQ Orbitrap Velos was operated in data-dependent acquisition mode with the XCalibur software (version 2.0 SR2, Thermo Fisher Scientific). The survey scan MS was performed in the Orbitrap on the 350–1800 m/z mass range with the resolution set to a value of 60,000. The 20 most intense ions per survey scan were selected with an isolation width of 2 m/z for subsequent data-dependent CID frag-mentation, and the resulting fragments were analyzed in the linear trap (LTQ). The normalized collision energy was set to 30%. To prevent repetitive selection of the same peptide, the dynamic exclusion duration was set to 60 s with a 10 ppm tolerance around the selected precursor and its isotopes. Monoisotopic precursor selection was turned on. For internal calibration the ion at 445.120025 m/z was used as lock mass.

## MS-based protein identification

Acquired MS and MS/MS data as raw MS files were converted to the mzDB format using the pwiz-mzdb converter (version 0.9.10, https://github.com/mzdb/pwiz-mzdb) executed with its default parameters (*Bouyssié et al., 2015*). Generated mzDB files were processed with the mzdb-access library (version 0.7, https://github.com/mzdb/mzdb-access; *Bouyssié et al., 2020*) to generate peak lists. Peak lists were searched against the UniProtKB/Swiss-Prot protein database with *Mus musculus* taxonomy (16,979 sequences) in the Mascot search engine (version 2.6.2, Matrix Science, London, UK). Cysteine carbamidomethylation was set as a fixed modification and methionine oxidation as a variable modification. Up to two missed trypsin/P cleavages were allowed. Mass tolerances in MS and MS/MS were set to 10 ppm and 0.6Da, respectively. Validation of identifications was performed through a false-discovery rate set to 1% at protein and peptide-sequence match level, determined by target-decoy search using the in-house-developed software Proline software version 1.6 (*Bouyssié et al., 2020*).

## Polysomal RNA preparation

HL-1 cells were cultured in 150 mm dishes. Ten0 min before harvesting, cells were treated with cyclo-heximide at 100 mg/mL. Cells were washed with PBS at room temperature containing 100 mg/mL cycloheximide and harvested with Trypsin. After centrifugation at 500 g for 3 min at 4 °C, cells were washed two times in PBS cold containing 100 mg/mL cycloheximide, and cells were lysed by hypo-tonic lysis buffer (10 mM HEPES-KOH Ph7.5; 10 mM KCl; 1.5 mM MgCl$_2$) containing 100 mg/mL cyclo-heximide. Cells were centrifuged at 500 g for 3 min and lysed by lysis solution containing hypotonic buffer, 1 mM DTT, 0.5 U/mL Rnasin, and protease inhibitor 100 X. Cells were centrifuged by two times, first at 1000 g for 10 min at 4 °C and second at 10,000 g for 15 min; the supernatants were collected and loaded onto a 10–50% sucrose gradient. The gradients were centrifuged in a Beckman SW41 Ti rotor at 39,000 rpm for 2.3 hr at 4 °C without a brake. Fractions were collected using a Foxy JR ISCO collector and UV optical unit type 11. RNA was purified from pooled heavy fractions containing poly-somes (fractions 12–19) as well as from cell lysate before gradient loading.

## qPCR fluidigm array

The DELTAgene Assay was designed by Fluidigm Corporation (San Francisco, USA). The qPCR-array was performed on BioMark with the Fluidigm 96.96 Dynamic Array following the manufactur-er's protocol (Real-Time PCR Analysis User Guide PN 68000088). The list of primers is provided in *Supplementary file 2*. A total of 25 ng of cDNA was preamplified using PreAmp Master Mix (Flui-digm,100–5581, San Francisco, USA) in the plate thermal cycler at 95 °C for 10 min, 16 cycles at 95 °C for 15 sec, and 60 °C for 4 min. The preamplified cDNA was treated with Exonuclease I in the plate thermal cycler at 37 °C for 30 min, 80 °C for 15 min and 10 °C infinity. The preamplified cDNA was mixed with 2 x TaqMan Gene Expression Master Mix (Applied Biosystems), 20 µM of mixed forward and reverse primers, and sample Loading Reagent (Fluidigm, San Francisco, USA). The sample was loaded into the Dynamic Array gene expression 96.96 IFC (Fluidigm San Francisco, USA). The qPCR

reactions were performed in the BioMark RT-qPCR system. Data were analyzed using the BioMark RT-qPCR Analysis Software Version 4.5.2.

GAPDH rRNA was used as a reference gene, and all data were normalized based on GAPDH rRNA level. Relative quantification (RQ) of gene expression was calculated using the $2^{-\Delta\Delta CT}$ method. When the RQ value was inferior to 1, the fold change was expressed as $-1/RQ$. The oligonucleotide primers used are detailed in *Supplementary file 2*.

## Quantification and statistical analysis qPCR and ddPCR analysis

qPCR data were analyzed on Quantstudio (AppliedBiosystems). RPL11 or HPRT were used as reference gene. Relative quantification (RQ) of gene expression was calculated using the $2-\Delta\Delta CT$ method. ddPCR data was analyzed using the QuantaSoft 1.7.4 software (Bio-Rad). HPRT was used as a reference gene, and Neat1 RNA expression was normalized by normoxia control and expressed in %.

### Label-free quantitative proteomics analysis

For label-free relative quantification across samples, raw MS signal extraction of identified peptides was performed using Proline. The cross-assignment of MS/MS information between runs was enabled (it allows to assign peptide sequences to detected but non-identified features). Each protein intensity was based on the sum of unique peptide intensities and was normalized across all samples by the median intensity. Missing values were independently replaced for each run by its 5% quantile. After log2-transformation of the data, the values of the technical replicates were averaged for each analyzed samples. For each pairwise comparison, an unpaired two-tailed Student's t-test was performed. Proteins were considered significantly enriched when their absolute log2-transformed fold change was higher than 1 and their p-value lower than 0.05. To eliminate false-positive hits from quantitation of low-intensity signals, two additional criteria were applied: only the proteins identified with a total number of averaged peptide spectrum match (PSM) counts >4 and quantified in a minimum of two biological replicates, before missing value replacement, for at least one of the two compared conditions were selected. Volcano plots were drawn to visualize significant protein abundance variations between the two compared conditions. They represent -log10 (p-value) according to the log2 ratio. The complete list of proteins identified and quantified in immunopurified samples and analyzed according to this statistical procedure is described in *Supplementary file 7*.

### Dual luciferase system

Data were analyzed on MicroWin 2000. Background noise was measured with non-transduced cell samples and removed from transduced cell sample measurement. Then LucF/LucR ratio was calculated on Excel 2007 (Microsoft Office) and mean and SD were calculated as well.

### FISH

Images were analyzed with a script for ImageJ. For each segmented nucleus, spots are segmented by detecting local maxima after applying a laplacien filter. Spot colocalization is determined by the distance between them.

### Capillary Western

Data were analyzed on compass software provided by the manufacturer.

### Statistical analysis

All statistical analyses were performed using One Way ANOVA, unpaired two-tailed student t-test, or Mann-Whitney rank comparisons test calculated on GraphPad Prism software depending on n number obtained and experiment configuration. Results are expressed as mean ± standard deviation, *p<0.05, **p<0.01,***<0.001, ****<0.0001.

## Acknowledgements

Our thanks go to JJ Maoret and F Martins from the Inserm UMR1297 I2MC GeT-TQ plateau of the GeT platform Genotoul (Toulouse), A Lucas from the I2MC We-Met Functional Biochemistry Facility (Toulouse) and R Flores-Flores from the I2MC imaging plateau. We also thank L Colras for technical

assistance. We also thank P-E Gleizes and H Prats for helpful discussion. This work was supported by Région Occitanie (Midi-Pyrénées), Association pour la Recherche sur le Cancer (ARC), Fondation Toulouse Cancer Santé and Agence Nationale de la Recherche ANR-18-CE11-0020-RIBOCARD, European funds (Fonds Européens de Développement Régional, FEDER), Toulouse Métropole, and by the French Ministry of Research with the Investissement d'Avenir Infrastructures Nationales en Biologie et Santé program (ProFI, Proteomics French Infrastructure project, ANR-10-INBS-08). ACG, FH, and ER had fellowships from the Ligue Nationale Contre le Cancer (LNCC).

## Additional information

### Funding

| Funder | Grant reference number | Author |
| --- | --- | --- |
| Agence Nationale de la Recherche | ANR-18-CE11-0020-RIBOCARD | Anne-Catherine Prats |
| Agence Nationale de la Recherche | ProFI ANR-10-INBS-08 | Odile Burlet-Schiltz<br>Carine Froment |
| Ligue Contre le Cancer | | Anne-Claire Godet<br>Fransky Hantelys<br>Emilie Roussel |

The funders had no role in study design, data collection and interpretation, or the decision to submit the work for publication.

### Author contributions

Anne-Claire Godet, Conceptualization, Formal analysis, Validation, Investigation, Methodology, Writing - original draft, Writing - review and editing; Emilie Roussel, Conceptualization, Formal analysis, Investigation, Methodology, Writing - review and editing; Florian David, Conceptualization, Data curation, Formal analysis, Methodology; Fransky Hantelys, Florence Tatin, Formal analysis, Investigation, Methodology, Writing - review and editing; Florent Morfoisse, Anthony K Henras, Conceptualization, Formal analysis, Methodology, Writing - review and editing; Joffrey Alves, Formal analysis, Investigation, Methodology; Françoise Pujol, Investigation, Methodology; Isabelle Ader, Conceptualization, Formal analysis, Supervision, Writing - review and editing; Edouard Bertrand, Eric Lacazette, Conceptualization, Supervision, Methodology, Writing - review and editing; Odile Burlet-Schiltz, Formal analysis, Methodology, Writing - review and editing; Carine Froment, Conceptualization, Data curation, Formal analysis, Methodology, Writing - review and editing; Patrice Vitali, Conceptualization, Supervision, Methodology; Barbara Garmy-Susini, Conceptualization, Formal analysis, Supervision, Funding acquisition, Writing - review and editing; Anne-Catherine Prats, Conceptualization, Formal analysis, Supervision, Funding acquisition, Validation, Investigation, Methodology, Writing - original draft, Project administration, Writing - review and editing

### Author ORCIDs

Florian David (ID) http://orcid.org/0000-0001-9842-1548
Carine Froment (ID) http://orcid.org/0000-0003-3688-5560
Anne-Catherine Prats (ID) http://orcid.org/0000-0002-5282-3776

### Decision letter and Author response

Decision letter https://doi.org/10.7554/eLife.69162.sa1
Author response https://doi.org/10.7554/eLife.69162.sa2

## Additional files

### Supplementary files

• Supplementary file 1. IRES activities in normoxic and hypoxic HL-1 cells after *Neat1* knock-down. HL-1 cells were transduced with Lucky Luke bicistronic lentivector containing the IRES of *FGF1* mRNA. Cells were submitted to normoxia or hypoxia 1% $O_2$ during 4 hr, 8 hr or 24 hr. Renilla and

firefly luciferase activities were measured (page 1) and the IRES activities evaluated with the ratio LucF/LucR (page 2). Mann-Whitney test was performed with n=12. **$P<0.01$. For each hypoxia condition the mean of the LucF/LucR ratio has been calculated with 12 cell culture biological replicates, normalized to normoxia Experiments A, B, C correspond to experiments performed at different dates, while 1, 2, 3, 4 are experiments performed in parallel at the same date, each of them being already the mean of three technical replicates (36 technical replicates in total).

- Supplementary file 2. Sequences of FISH and smiFISH probes, PCR primers, gapmers, siRNAs and Fluidigm probes. A/ FISH and smiFISH probes. B/ qPCR primers and cloning primers. C/ LNA gapmers used in *Neat1* knock-down assays. D/ SiRNA SMARTpools (Dharmacon) used for IRES activity studies. E/ Fluidigm deltagene primers. The corresponding genes are indicated in the left column.

- Supplementary file 3. IRES activities in normoxic and hypoxic HL-1 cells after *Neat1* knock-down. HL-1 cells were transduced with Lucky Luke bicistronic lentivectors containing the IRES of *FGF1*, *FGF2*, *VEGFA* (IRES a or b), *VEGFC*, *VEGFD*, *IGF1R*, *MYC* or EMCV. Cells were then treated with a pool of gapmers Neat1, or control during normoxia or hypoxia 1% $O_2$. For the lentivector with FGF1 IRES, cells were also treated with the gapmer Neat1_2 (N1_2, page 2). Renilla and firefly luciferase activities were measured (upper panel) and the IRES activities evaluated with the ratio LucF/LucR (lower panel). Mann-Whitney test was performed with n=9. *$P<0.05$, **$P<0.01$, ***$<0.001$, ****$P<0.0001$. For each IRES the mean has been calculated with nine cell culture biological replicates Experiments A, B, C correspond to experiments performed at different dates, while 1, 2, 3 are experiments performed in parallel at the same date, each of them being already the mean of three technical replicates (27 technical replicates in total).

- Supplementary file 4. *FGF1* IRES activity in normoxic and hypoxic HL-1 cells after *Sfpq, Rps2, Hnrnpm* knock-down. HL-1 cells were transduced with a Lucky Luke bicistronic lentivector containing the IRES of FGF1. Cells were then treated with siSfpq (A), siRps2 (B), sihnRNPM (C), siNucleolin (siNcl, D) or siControl (siCtrl) smartpools during normoxia or hypoxia 1% $O_2$. Renilla and firefly luciferase activities were measured and the IRES activities evaluated with the ratio LucF/LucR. Mann-Whitney test was performed with n=9 (n=12 for FGF1 IRES). *$P<0.05$, **$P<0.01$, ***$<0.001$, ****$P<0.0001$. For each siRNA the mean has been calculated with nine cell culture biological replicates Experiments A, B, C correspond to experiments performed at different dates, while 1, 2, 3 are experiments performed in parallel at the same date, each of them being already the mean of three technical replicates (27 technical replicates in total).

- Supplementary file 5. IRES activities in normoxic and hypoxic HL-1 cells after p54$^{nrb}$ knock-down. HL-1 cells were transduced with Lucky Luke bicistronic lentivectors containing the IRES of *FGF1*, *FGF2*, *VEGFA* (IRES a or b), *VEGFC*, *VEGFD*, *IGF1R*, *MYC* or EMCV. Cells were then treated with siP54$^{nrb}$ or siControl (siCtrl) smartpool during normoxia or hypoxia 1% $O_2$. Renilla and firefly luciferase activities were measured (upper panel, or page 1 for *FGF1* IRES) and the IRES activities evaluated with the ratio LucF/LucR (lower panel, or page 2 for *FGF1* IRES). Mann-Whitney test was performed with n=9 (n=12 for *FGF1* IRES). *$P<0.05$, **$P<0.01$, ***$<0.001$, ****$P<0.0001$. For each IRES the mean has been calculated with nine cell culture biological replicates Experiments A, B, C correspond to experiments performed at different dates, while 1, 2, 3 are experiments performed in parallel at the same date, each of them being already the mean of three technical replicates (27 technical replicates in total).

- Supplementary file 6. IRES activities in normoxic and hypoxic HL-1 cells after *Pspc1* knock-down. HL-1 cells were transduced with Lucky Luke bicistronic lentivectors containing the IRES of *FGF1*, *FGF2*, *VEGFA* (IRES a or b), *VEGFC*, *VEGFD*, *IGF1R*, *MYC* or EMCV. Cells were then treated with siPSPC1 or siControl (siCtrl) smartpool during normoxia or hypoxia 1% $O_2$. Renilla and firefly luciferase activities were measured (upper panel, or page 1 for *FGF1* IRES) and the IRES activities evaluated with the ratio LucF/LucR (lower panel, or page 2 for FGF1 IRES). Mann-Whitney test was performed with n=9 (n=12 for *FGF1* IRES). *$P<0.05$, **$P<0.01$, ***$<0.001$, ****$P<0.0001$. For each IRES the mean has been calculated with nine cell culture biological replicates Experiments A, B, C correspond to experiments performed at different dates, while 1, 2, 3 are experiments performed in parallel at the same date, each of them being already the mean of three technical replicates (27 technical replicates in total).

- Supplementary file 7. Label-free quantitative analysis of HA-P54$^{nrb}$-bound proteins identified by mass spectrometry in different conditions. p54$^{nrb}$-HA transduced HL-1 cells were subjected to normoxia or hypoxia, then nucleus and cytoplasm fractionation was performed and extracts were immunoprecipitated using anti-HA antibody. Enriched interacting proteins were identified by using a label-free quantitative mass spectrometry approach. The dataset list of proteins identified by MS/MS

is presented in alphabetical order. An unpaired bilateral student t-test with equal variance was used. Enrichment significance thresholds correspond to an absolute log2-transformed fold-change (FC) greater than 1 and a -log10-transformed (p-value) greater than 1.3.

• Supplementary file 8. Change of mRNA recruitment into polysomes following *Neat1* or *Neat1_2* knock-down. HL-1 cardiomyocytes were transfected with gapmer Neat1, Neat1_2, or control. Polysomes were purified on sucrose gradient as described in Star Methods. RNAs were purified from cytoplasmic extracts and from pooled polysomal fractions and analyzed on a Fluidigm deltagene PCR array from two biologicals replicates (cell culture dishes and cDNAs), each of them measured in three technical replicates (PCR reactions). mRNA levels in polysomes (polysomal RNA/ total RNA) were analyzed for each gene. Relative quantification (RQ) of mRNA level was calculated using the 2–ΔΔCT method with normalization to *Gapdh* mRNA and to HL-1 tranfected by gapmer control, and is shown as fold change of expression. RQ1 and RQ2 correspond to two independent experiments. To measure the fold change of repression, the mean is expressed as –1/RQ for the values <1 (yellow column).

• Transparent reporting form

### Data availability

Lentivector plasmid sequences are available on Dryad. https://doi.org/10.5061/dryad.2330r1b and https://doi.org/10.5061/dryad.m0cfxpp75. The MS proteomics data have been deposited to the ProteomeXchange Consortium via the PRIDE partner repository with the dataset identifier PXD024067.

The following datasets were generated:

| Author(s) | Year | Dataset title | Dataset URL | Database and Identifier |
|---|---|---|---|---|
| Froment C | 2021 | Long non-coding RNA Neat1 is a key translational regulator in hypoxia | http://proteomecentral.proteomexchange.org/cgi/GetDataset?ID=PXD024067 | ProteomeXchange, PXD024067 |
| Godet A, Roussel E, David FP, Hantelys F, Morfoisse F, Alves J, Pujol F, Ader I, Bertrand E, Burlet-Schiltz O, Froment C, Henras A, Vitali P, Lacazette E, Tatin F, Garmy-Susini B | 2022 | Long non-coding RNA Neat1 and paraspeckle components are translational regulators in hypoxia | https://doi.org/10.5061/dryad.m0cfxpp75 | Dryad Digital Repository, 10.5061/dryad.m0cfxpp75 |

The following previously published dataset was used:

| Author(s) | Year | Dataset title | Dataset URL | Database and Identifier |
|---|---|---|---|---|
| Hantelys F, Godet A, David F, Tatin F, Renaud-Gabardos E, Pujol F, Diallo L, Ader I, Ligat L, Henras A, Sato Y, Parini A, Lacazette E, Garmy-Susini B, Prats A | 2020 | Data from: Vasohibin1, a new IRES trans-acting factor for induction of (lymph)angiogenic factors in early hypoxia | https://doi.org/10.5061/dryad.2330r1b | Dryad Digital Repository, 10.5061/dryad.2330r1b |

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
