## [Editor Report]

The paper reports that the long non-coding RNA *Neat1* (nuclear paraspeckle assembly transcript 1) is required for IRES Internal Ribosome Entry Site)-mediated mRNA translation activity. *Neat1* is required for the activity of many cellular IRESs during the stress response in angiogenesis and/or cardio-protection. The authors conclude that nuclear paraspeckles serve as areas where cellular IRESes acquire ITAFs (IRES trans-activating factors. The findings of this paper have practical implications beyond a single subfield and the methods, data, and analyses broadly support the claims with only minor weaknesses.

---

## [Decision Letter]

**Decision letter after peer review:**

Thank you for submitting your article "Long non-coding RNA Neat1 is a key translational regulator in hypoxia" for consideration by *eLife*. Your article has been reviewed by 3 peer reviewers, and the evaluation has been overseen by a Reviewing Editor and James Manley as the Senior Editor. The reviewers have opted to remain anonymous.

The paper reports highly novel data that connect nuclear paraspeckles and lncRNA to IRES activity. However, many of the experiments are incomplete, and some of the conclusions are unjustified. Technical problems. such as missing loading controls must be rectified. The full reviewers' reports are included. They include a long list of criticisms and constructive suggestions for improvements.

Essential revisions:

1. Depletion of NEAT1 by control gapmers already exerts stress on the cardiomyocytes. Can the authors be sure that observed changes in IRES activities were not caused by higher levels of cell stress? Is eIF2alpha phosphorylated in the gapmer experiments (Figure 2)? If this is the case, it could explain why only a few differences were noted in Figure 4D.

Relevant to this point, you silence the long non-coding RNA, but transfection per se causes stress to cell. It seems that a different approach instead of silencing should be used since stress response is the question that needs to be answered. Perhaps a stable cell line or CRISPR KO can be used instead? Otherwise, the authors need to explain why the cell stress induced by transfection is not relevant here.

2. While NEAT1 abundance has an effect on IRES, the NEAT-ITAFs do not. The authors should check the effects of DDX17, Eno3, and Hspa2 that were enriched in the cytoplasm during hypoxia (Figure 4E).

3. Depletion of NEAT1 or NEAT1-ITAFs has always more effect on endogenous FGF1 protein abundance than on IRES activity. Is the half-life of FGF1 affected during the siRNA and gapmer treatments?

4. You need to provide a more direct cause-effect relationship between Neat1 and the proteins claimed to be ITAFs for the FGF1 IRES activity.

5. Validate the idea that paraspeckles are platforms for IRESome formation as this would imply the IRES needs to transit through the paraspeckle to gain full activity.

6. Validate some of the MS-IP data where 2013 proteins interacti with p54nrb.

7. There is an issue with the timing of the response, inasmuch as the amount of Neat1 induction of the FGF IRES and paraspeckle activity does not seem to align. Therefore, how can the conclusion be drawn that Neat1 is absolutely crucial?

8. Only one cell line was used in these studies. The authors make the point in the Discussion that IRESs can behave differently in different cell lines, which raises the question regarding how it can be known that Neat1 is generally important.

9. In the hypoxia studies, typically the rate of canonical protein synthesis is significantly reduced, mTOR is inactivated and eIF2α is phosphorylated. Does this occur in HL-1 cardiomyocytes under their conditions of early hypoxia and if so is mTOR inhibition or eIF2 phosphorylation necessary for the effects that they see? In other words, are the novel data presented connected to established effects of hypoxia on protein synthesis?

10. According to the data in Figure 1 and supplemental Figure 1, there is a better correlation with the formation of paraspeckles and greater expression under hypoxia of the longer isoform, NEAT1_2. You should explain the reason for focusing on just NEAT1, the shorter isoform, on the effect of the regulation of FGF1 IRES activity in Figure 2?

11. The mRNAs encoding the major (lymph)angiogenic growth factors are fibroblast growth factor (FGF) and vascular endothelial growth factor (VEGF). You should evaluate the effect of Neat1 on the regulation of more than just one IRES. The same experiments shown in Figures 1, 2, and 3 should be included for VEGF, especially since the results presently cannot be extrapolated to other IRES elements, and because recent studies have shown that FGF can act synergistically with VEGF.

*Reviewer #1 (Recommendations for the authors):*

The manuscript investigates an interesting topic regarding IRES regulation by the Neat1 lncRNA and several ITAFs. General conclusion regarding paraspeckles being sites of IRESome formation appear premature as they are supported by correlative information. It is difficult to conclude direct cause-effect relationships on a quick-response regulatory mechanism such as translation through the use of gene knockdowns. The reason being that it takes several days for appropriate knockdown levels of RNA or protein to be reached – making it difficult to eliminate the possibility that secondary/pleiotropic effects are responsible for, or significantly contributing to, the observed physiological effect.

I think experiments are required that (i) provide a more direct cause-effect relationship between Neat1 and the proteins claimed to be ITAFs for the FGF1 IRES activity, (ii) validate the idea that paraspeckles are platforms for IRESome formation as this would imply the IRES needs to transit through the paraspeckle to gain full activity, and (iii) validate the MS-IP data where 2013 proteins interacting with p54nrb.

Specific:

1. Section entitled: "FGF1 IRES activation during hypoxia correlates with paraspeckle formation and with Neat1 induction."

– Measurements from the RLuc/FLuc reporter are made 4, 8, 24 after induction of hypoxia. Please show individual RLuc and FLuc measurements, not the ratios. Is eIF2-α phosphorylated under the experimental conditions? Under hypoxic conditions, are the same isoforms of Neat1 produced to the same ratios? Are new isoforms seen? A Northern blot would provide insight into whether the architecture of this lncRNA changes.

– Last sentence of this paragraph reads "This correlation fits better with Neat1_2 than with the total Neat1, suggesting a link between expression of the long Neat1 isoform and IRES activation." Can correlation of Neat1 expression and IRES activation be used to imply a link between the two?

2. Section entitled: "LncRNA Neat1 knock-down drastically affects the FGF1 IRES activity and endogenous FGF1 expression."

The authors deplete Neat1 using gap-mers, I could not find how long a treatment was used to achieve the 3-fold knockdown (I'm assuming at least 2 days). The long time period in these experiments likely makes it difficult to conclude that reductions in Neat are directly responsible for the effects on FGF1 IRES activity or FGF1 protein levels. How selective is the reported effect for the FGF1 5' leader? Hence, I think it is premature to conclude that "Neat1 regulates FGF1 expression, and acts as an ITAF of the FGF1 IRES" (last sentence of paragraph) since an ITAF would imply a direct interaction (which is not shown in this MS).

3. Section entitled "Paraspeckle proteins P54nrb and PSCP1, but not SFPQ, are ITAFs of the FGF1 IRES." The authors assess three major components of paraspeckles (the SFPQ, p54nrb and PSPC1 proteins) for a potential role in IRES activity. Since knockdowns are used to reduce protein levels and this takes several days to achieve, how can effects on IRES activity or FGF1 protein levels be directly link to the protein activity targeted for knockdown versus a secondary effect arising due to target knockdown? How selective is the reported effect for the FGF1 5' leader? Hence, I think its difficult to conclude that "The ability of three paraspeckle components, Neat1, p54nrb and PSPC1 to regulate the FGF1 IRES activity led us to the hypothesis that the paraspeckle might be involved in the control of IRES-dependent translation. ". Can direct interaction be shown and can this, in turn, be linked to a translational response.

4. Section entitled "P54nrb interactome in normoxic and in hypoxic cardiomyocytes". The authors write "The moderate effect of p54nrb or PSPC1 depletion on FGF1 IRES activity suggested that other proteins are involved". Maybe the moderate effect is due to the poor knockdown achieved with the siRNAs (Figure 3 – Suppl Figure 1; 50% for p54nrb and 25% for PSPC1)? The IP performed with HA-tagged p54nrb pulled down 2013 proteins if I understand this section correctly. Can the authors explain why so many proteins are being pulled down? The supplemental table provided with all the raw data is not reader friendly and the column labeling difficult to interpret for someone like me who is not familiar with seeing data in this format. The identification of some ITAFs in this dataset is taken as evidence that these ITAFs interact with p54nrb in paraspeckles. What independent validation has been undertaken to show that these are associated with p54nrb? How many proteins that are not ITAFs are pulled down by p54nrb and enriched when comparing hypoxic versus normoxic extracts? In Figure 4D and 4E, the identity of all proteins enriched under the conditions shown should be shown in the upper rectangles.

5. Section entitled "Neat1 is the key activator of (lymph)angiogenic and cardioprotective factor mRNA IRESs." In this section, the authors test the response of other IRESes to knockdown of p54, PSPC1, and Neat1 and depending on the response, stratify the IRESes into different regulons based on IRESome composition. Whereas, the experiments do allow the authors to classify the response of the different IRESes, I don't think it provides insight into IRESome composition since the experiments to not show that any of the three tested molecules directly associate with the IRES under study.

*Reviewer #2 (Recommendations for the authors):*

1. There is an issue with timing of the response, the amount of Neat1 induction of the FGF IRES and paraspeckle activity that do not seem to align. Therefore, how can the conclusion be drawn that Neat1 is absolutely crucial.

2. The authors silence the long non-coding RNA but their transfection per se causes stress to cell. It seems that a different approach instead of silencing should be used since stress response is the question that needs to be answered. Perhaps a stable cell line or CRISPR KO can be used instead? Otherwise, the authors need to explain why the cell stress induced by transfection is not relevant here.

3. Only one cell line was used in these studies. The authors make the point in the Discussion that IRESs can behave differently in different cell lines, which raises the question regarding how it can be known that Neat1 is generally important.

4. How did the authors confirm HL-1 cardiomyocytes were indeed under hypoxia? Which is the hypoxia control? This is not apparent.

5. In the hypoxia studies, typically the rate of canonical protein synthesis is significantly reduced, mTOR is inactivated and eIF2α is phosphorylated. Does this occur in HL-1 cardiomyocytes under their conditions of early hypoxia and if so is mTOR inhibition or eIF2 phosphorylation necessary for the effects that they see? In other words, are the novel data presented connected to established effects of hypoxia on protein synthesis?

6. According to their data (Figure 1 and supplemental Figure 1), there is a better correlation with the formation of paraspeckles and greater expression under hypoxia of the longer isoform, NEAT1_2. Can the authors explain the reason for focusing on just NEAT1, the shorter isoform, on the effect of the regulation of FGF1 IRES activity in Figure 2?

7. The mRNAs encoding the major (lymph)angiogenic growth factors are fibroblast growth factor (FGF) and vascular endothelial growth factor (VEGF). The authors should evaluate the effect of Neat1 on the regulation of more than just one IRES. The same experiments shown in Figures 1, 2, and 3 should be included for VEGF, especially since the results presently cannot be extrapolated to other IRES elements, and because recent studies have shown that FGF can act synergistically with VEGF.

8. If possible, the authors should provide immunoprecipitation and immunoblot analysis validation of their proteomic data.

9. Is the interaction between nucleolin and ribosomal protein Rps2 with p54 direct or indirect? The graphical abstract shows a direct interaction but there is no confirming evidence for this. Also, does this potential interaction only occur under hypoxia?

10. Figures 1-3 need a protein loading control, perhaps actin.

11. Can the authors show that levels of FGF1 protein after SFPQ knockdown are unchanged?

*Reviewer #3 (Recommendations for the authors):*

My main concern is that depletion of NEAT1 by control gapmers already exerts stress on the cardiomyocytes. Can the authors be sure that observed changes in IRES activities were not caused by a higher levels of cell stress? Is eIF2alpha phosphorylated in the gapmer experiments (Figure 2)? If this is the case, it could explain why only few differences were noted in Figure 4D.

1. While NEAT1 abundance has an effect on IRES, the NEAT-ITAFs do not. The authors should check effects of DDX17, Eno3 and Hspa2 that were enriched in the cytoplasm during hypoxia (Figure 4E).

2. All figures displaying FISH results are impossible to inspect. They need to be bigger.

3. Depletion of NEAT1 or NEAT1-ITAFs has always more effect on endogenous FGF1 protein abundance that on IRES activity. Is the half-life of FGF1 affected during the siRNA and gapmer treatments?

[Editors' note: further revisions were suggested prior to acceptance, as described below.]

Thank you for resubmitting your work entitled "Long non-coding RNA Neat1 and paraspeckle components are translational regulators in hypoxia" for further consideration by *eLife*. Your revised article has been evaluated by James Manley (Senior Editor), and the Reviewing Editor and reviewers of the original manuscript.

The manuscript has been improved but there are some remaining issues that need to be addressed, as outlined below:

As detailed by reviewer #1 there are omissions in the text of the authors' responses listed in the rebuttal letter. Most importantly, since many of the results are correlative the conclusions are overstated. As requested by reviewer # 1, since the authors do not show a direct association between Neat1 and the ITAFs investigated herein, the authors ought to change or remove Figure 4A which claims that Neat1 and p54nrb interact with the FGF1 IRES.

*Reviewer #1 (Recommendations for the authors):*

The authors have provided new information which addresses some of my previous concerns. However, my main concern remains in that there is much correlative data from which direct cause-effect relationships are inferred.

The authors indicate that there was a decrease in the number of paraspeckles in cells in which Neat1 was knocked down, as well as a 2-fold reduction in IRES activity. However, in new data presented, the authors show that eIF2alpha is phosphorylated upon Neat1_2 KO (Figure 2, Suppl 4) and in their rebuttal write that this could explain the small difference of IRES activity between normoxia and hypoxia generated by the KO. I, therefore, don't understand how the authors conclude on line 232 of the MS that the Neat1 knockdown results suggest "that Neat1 regulates FGF1 mRNA translation, directly or indirectly". The same comments holds true for the data regarding changes in the polysomal mRNA context following Neat1 knockdown (Figure 8). Are these due to elevated eIF2 α phosphorylation and not necessarily due to the absence of Neat1

The authors use smiFISH to probe the localization of Neat 1 and FGF1 IRES and find a ~ 2-fold increase in overlapping signals for the FGF1 IRES and Neat1 in hypoxic versus normal cells. (Figure 3B). I'm not sure how this led to the "hypothesis that the paraspeckle might be involved in the control of IRES-dependent translation." For this conclusion to be made, wouldn't one have to show altered IRES activity after versus before having encountered the Neat1 speckles?

Since the authors do not show a direct association between Neat1 and the ITAFs investigated herein, I would ask that they change or remove Figure 4A which appears to show Neat1 and p54nrb interacting with the FGF1 IRES.

---

## [Author Response]

Essential revisions:1. Depletion of NEAT1 by control gapmers already exerts stress on the cardiomyocytes. Can the authors be sure that observed changes in IRES activities were not caused by higher levels of cell stress?

As written in page 5, 1^st^ paragraph, in the experiments with gapmer treatment the number of paraspeckles was already high in normoxia (almost 5 foci per cell). This may suggest suggesting that cells were already stressed by the gapmer treatment, before being submitted to hypoxia. Alternatively this higher number may result from the higher sensitivity of the smiFISH method compared to FISH in Figure 1.

Is eIF2alpha phosphorylated in the gapmer experiments (Figure 2)? If this is the case, it could explain why only a few differences were noted in Figure 4D.

eIF2a phosphorylation has been analyzed after gapmer treatment. (Figure 2, figure supplement 4). Our hypothesis to explain the effect of Neat1 depletion in normoxia is indeed that cells are stressed by the gapmer treatment. We performed the experiment (Figure 2, figure supplement 4): eIF2a phosphorylation slightly increases upon Neat1_2 gapmer treatment, but not using the control gapmer. This could explain the small difference of IRES activity between normoxia and hypoxia generated by the knock-down.

Relevant to this point, you silence the long non-coding RNA, but transfection per se causes stress to cell. It seems that a different approach instead of silencing should be used since stress response is the question that needs to be answered. Perhaps a stable cell line or CRISPR KO can be used instead?

Indeed the stress induced by transfection is an important limitation and it would have been interesting but this experiment. We tried to generate Neat1 CRISPR-based induction and silencing using lentivectors (with Cas9-VP160 and Cas-9-Krab, respectively, but we were unfortunately not successful). One gRNA was not sufficient, we obtain very low titers of the lentivector and in addition it was impossible in our hands to construct a lentivector coding several guides with the mCas9.

Otherwise, the authors need to explain why the cell stress induced by transfection is not relevant here.

I agree with the reviewer that the stress generated by transient transfection is a limitation of our approach. However the gapmer was the best tool to deplete Neat1, a nuclear ncRNA. As you can see in the Figure 1—figure supplement 3, we constructed lentivectors coding artificial miRNAs against Neat1 to deplete Neat1 in 67NR cells. This was done before the experiments with HL-1. We gave up this approach as it was very difficult to have an efficient knock-down. Gapmer provided better data. The stress due to the gapmer transfection can explain why we have no difference between normoxia and hypoxia, whereas we should have less effect in normoxia. Nevertheless the effect of Neat1 depletion on the activities of all IRESs except for EMCV IRES, remains a very convincing result by showing the role of Neat1 in IRES-dependent translation.

2. While NEAT1 abundance has an effect on IRES, the NEAT-ITAFs do not.

As mentioned in the text, the depletion of paraspeckle proteins had different effects depending on the IRES and we make the hypothesis that IRESs could be regrouped in different “regulons” in normoxia and in hypoxia (Figure 7D). We propose that IRESome composition varies for each IRES.

The authors should check the effects of DDX17, Eno3, and Hspa2 that were enriched in the cytoplasm during hypoxia (Figure 4E).

The knock-down of DDX17 has been performed. Unexpectedly, DDX17 knock-down has a slight positive effect on IRES activity in normoxia. We do not wish to publish these data here because the role of helicases is being addressed in our laboratory and will constitute another publication. However we include these data in the present letter.

**Author response image 1. sa2fig1:** FGF1 IRES activity after DDX17 knock-down.

3. Depletion of NEAT1 or NEAT1-ITAFs has always more effect on endogenous FGF1 protein abundance than on IRES activity. Is the half-life of FGF1 affected during the siRNA and gapmer treatments?

The half-life of FGF1 has been analyzed and does not vary upon gapmer treatment (Figure 2—figure supplement 5-6). It is superior to 24 hr either without treatment, or with gapmer treatment, or with siRNA treatment (control and target-specific). A hypothesis to explain a more important effect on endogenous FGF1 mRNA than bicistronic mRNA translation may be that the IRES of the endogenous FGF1 mRNA is recruited more efficiently to the paraspeckle than the IRES present in the bicistronic mRNA, whose structure may be altered due to the presence of the upstream cistron. However, if one looks at the Western quantification (normalized to total proteins), the difference is not so big: upon Neat1 depletion endogenous FGF1 protein is 34%, while IRES activity is 50% (Figure 2C and 2D). Upon p54 or PSPC1 depletion, endogenous FGF1 is 61% or 82% while IRES activity is 70% or 80%, respectively (Figure 4E and 4F).

4. You need to provide a more direct cause-effect relationship between Neat1 and the proteins claimed to be ITAFs for the FGF1 IRES activity.

The direct interaction of at least p54 and SFPQ with Neat1 has been established previously (for example see: Yamazaki et al. (2018). Functional Domains of NEAT1 Architectural lncRNA Induce Paraspeckle Assembly through Phase Separation. Molecular cell, 70(6), 1038–1053.e7. https://doi.org/10.1016/j.molcel.2018.05.019). The three DBHS proteins (p54, SFPQ and PSPC1) are known to function in heteroduplex (possible in vitro but not physiologically significant).

Furthermore, we show by smi-FISH in this revised version that Neat1 interacts with the IRES-containing mRNA (Figure 3).

As regards the other proteins, it is not necessary that interact directly with Neat1 to function as ITAFs, but we can propose that they are in the RNP complex assembled to the IRES in the paraspeckle (as they interact with p54^nrb^).

5. Validate the idea that paraspeckles are platforms for IRESome formation as this would imply the IRES needs to transit through the paraspeckle to gain full activity.

To validate the idea that the paraspeckle is a platform for IRESome assembly we performed an experiment of smiFISH (single molecule inexpensive FISH). This method using unlabeled primary probes and a fluorescently labelled secondary detector oligonucleotide allows to detect single RNA molecules (Tsanov et al. Nucleic Acids Res. 2016 Dec 15; 44(22): e165. 10.1093/nar/gkw784)*.* The results show that the bicistronic mRNA containing the IRES is significantly co-localized with Neat1, which indicates its recruitment in the paraspeckle (Figure 3). Furthermore, the presence of Neat1 in the cytoplasm, observed in our data (Figure 1, Supplement 2, and Figure 3), as well as its presence in hypoxic polysomes shown in our previous paper (Hantelys et al., *eLife* 2019), support the hypothesis of an export of this complex (including Neat) out of the nucleus.

6. Validate some of the MS-IP data where 2013 proteins interacti with p54nrb.

The interaction of nucleolin with p54^nrb^ has been validated by co-immunoprecipitation. (Figure 5—figure supplement 2).

A list of proteins present in the paraspeckle complex with p54nrb has been reported previously, in addition to the main paraspeckle proteins SFPQ and PSPC1 known to interact with p54 (Both of them are found in our MS-IP)(Figure 5—figure supplement 1, and MS data in Supplement file 7). Among 40 paraspeckle proteins listed in previous reports (Naganuma et al., EMBO J, 2012 and Yamamoto et al 2021), 22 have been found interacting with p54nrb in our SM-IP (supplementary file 7). Furthermore, among these 22 proteins, 6 have been described as ITAFs: FUS, hnRNPA1, hnRNPK, hnRNPM, p54nrb/NONO, SFPQ/PSF (Godet et al., IJMS 2018). This analysis has been added page 6, 1^st^ paragraph, and (Table 1). In our view it is significant to have found more than half of the main paraspeckle proteins in our MS-IP.

7. There is an issue with the timing of the response, inasmuch as the amount of Neat1 induction of the FGF IRES and paraspeckle activity does not seem to align. Therefore, how can the conclusion be drawn that Neat1 is absolutely crucial?

We agree that the correlation between IRES activation and Neat1 induction is not concordant (Figure 1), but it is fully concordant with Neat1_2: the IRES shows a peak of activation at 4 hr, whereas Neat1_2 induction and its presence in paraspeckles also exhibit a peak at 4 hr (observed Figure 1 with both RT qPCR and FISH). This correlation favors the hypothesis of an important role for Neat1_2, although our data with Neat1_2 gapmer had only a small impact on IRES activity (Figure 2 figure supplement 3).

8. Only one cell line was used in these studies. The authors make the point in the Discussion that IRESs can behave differently in different cell lines, which raises the question regarding how it can be known that Neat1 is generally important.

The experiments have been performed with another cell line, 67NR (mouse breast tumor Figure 1—figure supplement 3). Data show that this cell line is more resistant to hypoxia: Neat1 is not induced whereas Neat1_2 is strongly induced after 24 hr of hypoxia. This time, 24 hr, also corresponds to the IRES activation. Furthermore, in 67NR, we performed Neat1 and Neat1_2 knock-down: the data show that the role of Neat1 in the control of IRES-dependent translation is not limited to HL-1 cells.

9. In the hypoxia studies, typically the rate of canonical protein synthesis is significantly reduced, mTOR is inactivated and eIF2α is phosphorylated. Does this occur in HL-1 cardiomyocytes under their conditions of early hypoxia and if so is mTOR inhibition or eIF2 phosphorylation necessary for the effects that they see? In other words, are the novel data presented connected to established effects of hypoxia on protein synthesis?

4E-BP1 and eIF2a phosphorylation have been analyzed in these conditions of early hypoxia in HL-1 and published in our previous paper in *eLife* (Hantelys et al. 2019, https://elifesciences.org/articles/50094 Figure 2 and Figure 2 Supplement 4). Clearly, no change in 4E-BP phosphorylation is observed but eIF2a is phosphorylated in these conditions. We also showed in the paper by Hantelys *et al.* that global protein synthesis is decreased (P/M ratio, Figure 2).

10. According to the data in Figure 1 and supplemental Figure 1, there is a better correlation with the formation of paraspeckles and greater expression under hypoxia of the longer isoform, NEAT1_2. You should explain the reason for focusing on just NEAT1, the shorter isoform, on the effect of the regulation of FGF1 IRES activity in Figure 2?

The knock-down of Neat1_2 has been performed and shows a decrease of FGF1 IRES activity (Figure 2figure supplement 3). Unfortunately, it is not possible to knock-down only Neat1-1 as its sequence is entirely comprised in that of Neat1_2 (Figure 1E). These data, in addition to the results provided by fluidigm (Figure 8) and knock-down in 67NR cells (Figure 1—figure supplement 3) indicate that Neat1_2 is involved in the regulation. However, the knock-down of both isoforms (Neat1) has a more important effect on IRES activity, suggesting that both isoforms play a role. Due to the data with Neat1_2 gapmer (Figure 2—figure supplement 3) we have moderated the conclusion about the key role of Neat1_2 that we were not able to demonstrate.

11. The mRNAs encoding the major (lymph)angiogenic growth factors are fibroblast growth factor (FGF) and vascular endothelial growth factor (VEGF). You should evaluate the effect of Neat1 on the regulation of more than just one IRES. The same experiments shown in Figures 1, 2, and 3 should be included for VEGF, especially since the results presently cannot be extrapolated to other IRES elements, and because recent studies have shown that FGF can act synergistically with VEGF.

IRES activation in early hypoxia shown in Figure 1 has been published for several other IRESs of the FGF and VEGF families, in the same cell model and with the same bicistronic lentivectors, in our previous paper (Hantelys et al., *eLife* 2019).

Neat1, p54^nrb^ and PSPC1 knock-down shown in Figure 2 and 4 for FGF1 IRES have been assessed with the IRESs of FGF2, VEGFA, VEGFC, VEGFD, IGF1R, c-myc and EMCV in Figure 7 (Figure 6 in the first submission). Data show that Neat1 knock-down affects all IRESs except for EMCV, while knock-down of p54^nrb^ or PSPC1 only affects groups of IRESs.

Reviewer #1 (Recommendations for the authors):The manuscript investigates an interesting topic regarding IRES regulation by the Neat1 lncRNA and several ITAFs. General conclusion regarding paraspeckles being sites of IRESome formation appear premature as they are supported by correlative information. It is difficult to conclude direct cause-effect relationships on a quick-response regulatory mechanism such as translation through the use of gene knockdowns. The reason being that it takes several days for appropriate knockdown levels of RNA or protein to be reached – making it difficult to eliminate the possibility that secondary/pleiotropic effects are responsible for, or significantly contributing to, the observed physiological effect.I think experiments are required that (i) provide a more direct cause-effect relationship between Neat1 and the proteins claimed to be ITAFs for the FGF1 IRES activity, (ii) validate the idea that paraspeckles are platforms for IRESome formation as this would imply the IRES needs to transit through the paraspeckle to gain full activity, and (iii) validate the MS-IP data where 2013 proteins interacting with p54nrb.

To validate the idea that the paraspeckle is a platform for IRESome we performed an experiment of smiFISH (see point 5 above). The results show that the bicistronic mRNA containing the IRES is significantly co-localized with Neat1, which indicates its recruitment in the paraspeckle, supporting the hypothesis that paraspeckles are platforms of IRESome formation (Figure 3).

Also, the interaction of nucleolin with p54^nrb^ has been validated by co-immunoprecipitation (Figure 5—figure supplement 2).

A list of proteins present in the paraspeckle complex with p54^nrb^ has been reported previously, in addition to the main paraspeckle proteins SFPQ and PSPC1 known to interact with p54 (Both of them are found in our MS-IP) (Figure 5—figure supplement 1, and MS data in Supplement file 7). Among 40 paraspeckle proteins listed in previous reports (Naganuma et al., EMBO J, 2012 and Yamamoto et al. 2021), 22 have been found interacting with p54^nrb^ in our SM-IP (supplementary file 7). Furthermore, among these 22 proteins, 6 have been described as ITAFs: FUS, hnRNPA1, hnRNPK, hnRNPM, p54^nrb^/NONO, SFPQ/PSF (Godet et al., IJMS 2018). This analysis has been added page 12 and Table 1. In our view it is significant to have found more than half of the main paraspeckle proteins in our MSIP.

Specific:1. Section entitled: "FGF1 IRES activation during hypoxia correlates with paraspeckle formation and with Neat1 induction."– Measurements from the RLuc/FLuc reporter are made 4, 8, 24 after induction of hypoxia. Please show individual RLuc and FLuc measurements, not the ratios. Is eIF2-α phosphorylated under the experimental conditions? Under hypoxic conditions, are the same isoforms of Neat1 produced to the same ratios? Are new isoforms seen? A Northern blot would provide insight into whether the architecture of this lncRNA changes.

Indeed the table with LucR and LucF activities of Figure 1 data had been forgotten in the first submission. It is a mistake. The LucR and LucF values are now presented in Figure 1 Supplement 1 and new Supplementary file 1. eIF2a is already phosphorylated after 4 hours of hypoxia (published in our previous paper Hantelys et al., 2019, Figure 2 Supplement 4). Interestingly, LucR activity increases at 4 hr, and decreases only after 24 hr of hypoxia. This suggests an effect of the IRES on translation of the first cistron, as observed previously (Conte et al., PLoS One. 2008 Aug 27;3(8):e3078. doi: 10.1371/journal.pone.0003078.)

The experiments presented in Figure 1 were done in parallel to that published in 2019.

Regarding Neat1 isoforms, we can see in Figure 1D that the global Neat1 (two isoforms) induction is different from that of Neat 1-2 isoform. It is impossible to detect Neat1-1 shorter isoform separately by RT PCR as its sequence is contained in that of Neat1_2. The paraspeckle quantification confirms that difference in Figure 1F-K (FISH probing either Neat1 or Neat1_2): Neat1_2 shows a peak at 4 hr of hypoxia (as for RTqPCR), whereas for Neat1 reaches a plateau at 4 hr. This indicates a variation of the isoform ratio.

– Last sentence of this paragraph reads "This correlation fits better with Neat1_2 than with the total Neat1, suggesting a link between expression of the long Neat1 isoform and IRES activation." Can correlation of Neat1 expression and IRES activation be used to imply a link between the two?

We agree that a correlation is not a link but this observation incited us to look for a link, as p54 had been identified as an ITAF in our previous paper (Ainaoui et al., PlOS ONE 2015). The sentence has been changed (page 4, penultimate paragraph).

2. Section entitled: "LncRNA Neat1 knock-down drastically affects the FGF1 IRES activity and endogenous FGF1 expression."The authors deplete Neat1 using gap-mers, I could not find how long a treatment was used to achieve the 3-fold knockdown (I'm assuming at least 2 days). The long time period in these experiments likely makes it difficult to conclude that reductions in Neat are directly responsible for the effects on FGF1 IRES activity or FGF1 protein levels. How selective is the reported effect for the FGF1 5' leader? Hence, I think it is premature to conclude that "Neat1 regulates FGF1 expression, and acts as an ITAF of the FGF1 IRES" (last sentence of paragraph) since an ITAF would imply a direct interaction (which is not shown in this MS).

The times of gapmer treatment (48 hr for Neat 1 and 72 hr for Neat1_2 knock-down) are indicated in the “Cell transfection” section of Material and methods. However it has been added in the figure legends (Figure 2 and Figure 2 supplement 1). It is indeed premature to conclude that Neat1 is an ITAF at this stage. However the definition of ITAF does not mean that there is a direct interaction with the IRES: by example the ncRNA TMRP acts as a negative IRES trans-acting factor by competing for PTB binding and preventing it to bind to the IRES (Godet et al., IJMS 2019, Yang et al., Cell Death Diff 2018). By this way TRMP regulates p27^kip^ expression. In our view the IRESome is not limited to the protein directly linked to the mRNA but to a protein (or RNP) complex. Many ncRNAs are considered as regulators of gene expression without binding directly to mRNA or gene promoters: lcnRNAs and circRNAs are sponges for miRNAs and RBPs and by this mechanism they regulate gene expression. In our case Neat1, by participating in forming the IRESome by binding different RBPs present in the paraspeckle, acts as a positive regulator of gene expression. The smiFISH data showing the co-localization of Neat1 with the IRES-containing mRNA (Figure 3). The role of Neat1 in translational control is also supported by its export from the nucleus observed in Figure 1F-K, Figure 1—figure supplement 2 and Figure 3, and its presence in hypoxic polysomes shown in our previous paper (Hantelys et al., 2019, Figure 2).

3. Section entitled "Paraspeckle proteins P54nrb and PSCP1, but not SFPQ, are ITAFs of the FGF1 IRES." The authors assess three major components of paraspeckles (the SFPQ, p54nrb and PSPC1 proteins) for a potential role in IRES activity. Since knockdowns are used to reduce protein levels and this takes several days to achieve, how can effects on IRES activity or FGF1 protein levels be directly link to the protein activity targeted for knockdown versus a secondary effect arising due to target knockdown? How selective is the reported effect for the FGF1 5' leader? Hence, I think its difficult to conclude that "The ability of three paraspeckle components, Neat1, p54nrb and PSPC1 to regulate the FGF1 IRES activity led us to the hypothesis that the paraspeckle might be involved in the control of IRES-dependent translation. ". Can direct interaction be shown and can this, in turn, be linked to a translational response.

The direct interaction of at least p54 and SFPQ with Neat1 has been established previously, and the three DBHS proteins (p54, SFPQ and PSPC1) are known to function in heteroduplex (for example see: Yamazaki et al. (2018). Functional Domains of NEAT1 Architectural lncRNA Induce Paraspeckle Assembly through Phase Separation. Molecular cell, 70(6), 1038–1053.e7. https://doi.org/10.1016/j.molcel.2018.05.019). The interaction of Neat1 with the IRES-containing mRNA has been shown by smi-FISH (Figure 3). At this stage we cannot conclude whether the IRES interacts with Neat1 or with one or several of the paraspeckle proteins, but we can conclude that they are all in the same nuclear body. In a recent paper, it has been established that Neat1 interacts directly with several mRNA targets (Jacq et al., RNA Biol. 2021), thus we cannot exclude the hypothesis of a direct RNA-RNA interaction of the IRES with Neat1.

The selectivity for FGF1 leader (and other growth factor leaders) is shown in Figure 7: The EMCV IRES is not regulated by either of these paraspeckle components. The set of cellular IRESs assessed are all regulated by Neat1, and selectively by the paraspeckle proteins p54 and PSPC1.

4. Section entitled "P54nrb interactome in normoxic and in hypoxic cardiomyocytes". The authors write "The moderate effect of p54nrb or PSPC1 depletion on FGF1 IRES activity suggested that other proteins are involved". Maybe the moderate effect is due to the poor knockdown achieved with the siRNAs (Figure 3 – Suppl Figure 1; 50% for p54nrb and 25% for PSPC1)? The IP performed with HA-tagged p54nrb pulled down 2013 proteins if I understand this section correctly. Can the authors explain why so many proteins are being pulled down? The supplemental table provided with all the raw data is not reader friendly and the column labeling difficult to interpret for someone like me who is not familiar with seeing data in this format. The identification of some ITAFs in this dataset is taken as evidence that these ITAFs interact with p54nrb in paraspeckles. What independent validation has been undertaken to show that these are associated with p54nrb? How many proteins that are not ITAFs are pulled down by p54nrb and enriched when comparing hypoxic versus normoxic extracts? In Figure 4D and 4E, the identity of all proteins enriched under the conditions shown should be shown in the upper rectangles.

We agree that the moderate effect is due to the poor knock-down. HL1 are beating cardiomyocytes (see Video1), not efficiently transfected (and for that reason we have transduced them with lentivectors expressing the bicistronic reporter constructs). This has been changed in the text, page 6 2^nd^ paragraph. It was not expected to pull down so many proteins (Figure 5). This experiment has been done in independent quadruplicates. A possible explanation is that the paraspeckle is a huge RNP: a recent paper showed that 4268 RNAs are targeted to paraspeckle (Jacq et al., RNA Biol 2021). A single paraspeckle contains about 50 molecules of Neat 1-2 (Hirose WIR RNA 2019) binding multiple proteins. Many proteins are components of paraspeckle but the proteins interacting with Neat1 are mainly the core proteins p54 and SFPQ, thus is expected that p54 may form a big complex with many associated proteins. In addition, p54 can form complexes outside the paraspeckle (role in splicing).

Among 40 paraspeckle proteins listed in previous reports (Naganuma et al., EMBO J, 2012 and Yamamoto et al. 2021), 22 have been found interacting with p54^nrb^ in our SM-IP (supplementary file 7). Furthermore, among these 22 proteins, 6 have been described as ITAFs: FUS, hnRNPA1, hnRNPK, hnRNPM, p54^nrb^/NONO, SFPQ/PSF (Godet et al., IJMS 2018). This analysis has been added page 6 and Table 1. In our view it is significant to have found more than half of the main paraspeckle proteins in our MS-IP. The identity of all enriched proteins has been shown in Figure 5D and 5E.

5. Section entitled "Neat1 is the key activator of (lymph)angiogenic and cardioprotective factor mRNA IRESs." In this section, the authors test the response of other IRESes to knockdown of p54, PSPC1, and Neat1 and depending on the response, stratify the IRESes into different regulons based on IRESome composition. Whereas, the experiments do allow the authors to classify the response of the different IRESes, I don't think it provides insight into IRESome composition since the experiments to not show that any of the three tested molecules directly associate with the IRES under study.

We agree we have not definitely proven about IRESome composition by our data suggest it anyway, as we show that p54 interacts with PSPC1 and nucleolin (Figure 5), and we also show the colocalization of IRES-containing mRNA with Neat1 in the paraspeckle (Figure 3). Our data, and the direct interactions shown between PSP proteins and Neat1 in previous reports provide a set of arguments to support our proposition.

Reviewer #2 (Recommendations for the authors):1. There is an issue with timing of the response, the amount of Neat1 induction of the FGF IRES and paraspeckle activity that do not seem to align. Therefore, how can the conclusion be drawn that Neat1 is absolutely crucial.

The amount of Neat1 (Neat1-1 + Neat1_2) does not align indeed, but the amount of Neat1_2 shows a peak detected by RT PCR as well by FISH (Figure 1), which completely aligns with the FGF1 IRES peak of activity at 4h. In addition, this alignment is also observed in another cell line, the 67NR breast tumor murine cell line (Figure 1—figure supplement 3). Data show that this cell line is more resistant to hypoxia: Neat1 is not induced whereas Neat1_2 is strongly induced after 24h of hypoxia. This time, 24 hr, also corresponds to the IRES activation. Furthermore, in 67NR, we performed Neat1 and Neat1_2 knockdown, which generated IRES downregulation.

2. The authors silence the long non-coding RNA but their transfection per se causes stress to cell. It seems that a different approach instead of silencing should be used since stress response is the question that needs to be answered. Perhaps a stable cell line or CRISPR KO can be used instead? Otherwise, the authors need to explain why the cell stress induced by transfection is not relevant here.

We tried several knock-down approaches in HL-1 without success (siRNA, phosphorothiate oligonucleotides, shMir…). The gapmer was the best way to obtain a significant knock-down in these cells. We agree that CRISPR KO would be a nice approach and we also thought about it but it seemed us quite long to develop, as our experiments have already been delayed by the confinement and stock ruptures related to COVID19. We tried however to generate Neat1 CRISPR-based induction and silencing using lentivectors (with Cas9-VP160 and Cas-9-Krab, respectively, but we were unfortunately not successful). One gRNA was not sufficient, we obtain very low titers of the lentivector and in addition it was impossible in our hands to construct a lentivector coding several guides with the mCas9.

3. Only one cell line was used in these studies. The authors make the point in the Discussion that IRESs can behave differently in different cell lines, which raises the question regarding how it can be known that Neat1 is generally important.

As mentioned above, the experiments have been performed in another cell line, the 67NR breast tumor murine cell line (Figure 1 figure supplement 3). Data show that this cell line is more resistant to hypoxia: Neat1 is not induced whereas Neat1_2 is strongly induced after 24 hr of hypoxia. This time, 24 hr, also corresponds to the IRES activation. Furthermore, in 67NR, we performed Neat1 and Neat1_2 knockdown, which generated IRES downregulation. These data show that the role of Neat1 in IRESdependent translation, is not limited to HL-1 cells.

4. How did the authors confirm HL-1 cardiomyocytes were indeed under hypoxia? Which is the hypoxia control? This is not apparent.

The hypoxia conditions had been set up in our previous paper (Hantelys et al., *eLife* 2019). The 0_2_ pressure is measured in the incubator. Then expression a HIF-1 target, VEGFA, were checked by RT PCR. Figure 1 in Hantelys et al. shows that the transcriptional response occurs at 8 hours. eIF2a phosphorylation was measured by capillary Western (Hantelys Figure 2), showing that phosphorylation appears at 4 hr, before the transcriptional response via HIF. The experiments presented in Figure 1 of the present paper were done in parallel to that published in 2019.

5. In the hypoxia studies, typically the rate of canonical protein synthesis is significantly reduced, mTOR is inactivated and eIF2α is phosphorylated. Does this occur in HL-1 cardiomyocytes under their conditions of early hypoxia and if so is mTOR inhibition or eIF2 phosphorylation necessary for the effects that they see? In other words, are the novel data presented connected to established effects of hypoxia on protein synthesis?

4E-BP1 and eIF2a phosphorylation have been analyzed in these conditions of early hypoxia in HL-1 and published in our previous paper in *eLife* (Hantelys et al. 2019, https://elifesciences.org/articles/50094 Figure 2 and this paper Figure 2—figure supplement 4). Clearly, no change in 4E-BP phosphorylation is observed but eIF2a is phosphorylated in these conditions. We also showed in that paper that global protein synthesis is decreased (P/M ratio, Figure 2).

6. According to their data (Figure 1 and supplemental Figure 1), there is a better correlation with the formation of paraspeckles and greater expression under hypoxia of the longer isoform, NEAT1_2. Can the authors explain the reason for focusing on just NEAT1, the shorter isoform, on the effect of the regulation of FGF1 IRES activity in Figure 2?

The knock-down of Neat1_2 has been performed and shows a decrease of FGF1 IRES activity (Figure 2figure supplement 3). Unfortunately, it is not possible to knock-down only Neat1-1 as its sequence is entirely comprised in that of Neat1_2 (Figure 1E). These data, in addition to the results provided by fluidigm (Figure 8) and knock-down in 67NR cells (Figure 1—figure supplement 3) indicate that Neat1_2 is involved in the regulation. However the knock-down of both isoforms (Neat1) has a more important effect on IRES activity, suggesting that both isoforms play a role. Due to the data with Neat1_2 gapmer (Figure 2—figure supplement 3) we have moderated the conclusion about the key role of Neat1_2 that we were not able to demonstrate.

Please note that Neat1 corresponds to Neat1-1 + Neat1_2.

7. The mRNAs encoding the major (lymph)angiogenic growth factors are fibroblast growth factor (FGF) and vascular endothelial growth factor (VEGF). The authors should evaluate the effect of Neat1 on the regulation of more than just one IRES. The same experiments shown in Figures 1, 2, and 3 should be included for VEGF, especially since the results presently cannot be extrapolated to other IRES elements, and because recent studies have shown that FGF can act synergistically with VEGF.

It has been done in Figure 7: IRESs of FGF2, VEGFA, -C, D, IGF1R, c-myc and EMCV have been tested. Neat-1 appears as a key regulator for all IRESs except for EMCV IRES.

8. If possible, the authors should provide immunoprecipitation and immunoblot analysis validation of their proteomic data.

The interaction of p54 with nucleolin has been validated by co-immunoprecipitation. (Figure 5—figure supplement 2).

9. Is the interaction between nucleolin and ribosomal protein Rps2 with p54 direct or indirect? The graphical abstract shows a direct interaction but there is no confirming evidence for this. Also, does this potential interaction only occur under hypoxia?

By immunoprecipitation we show that p54 interacts with nucleolin Figure 5—figure supplement 2. Figure 5D shows that this interaction increases in hypoxia.

10. Figures 1-3 need a protein loading control, perhaps actin.

Capillary Western (Jess) does not need a loading control : quantification is obtained from peak area, than normalized to total proteins loaded on each capillary. It is much more precise than classical Western blots.

11. Can the authors show that levels of FGF1 protein after SFPQ knockdown are unchanged?

Unfortunately we did not measure endogenous FGF1 because the IRES activity did not change.

Reviewer #3 (Recommendations for the authors):My main concern is that depletion of NEAT1 by control gapmers already exerts stress on the cardiomyocytes. Can the authors be sure that observed changes in IRES activities were not caused by a higher levels of cell stress? Is eIF2alpha phosphorylated in the gapmer experiments (Figure 2)? If this is the case, it could explain why only few differences were noted in Figure 4D.

As written in page 8, 1^st^ paragraph, in the experiments with gapmer treatment the number of paraspeckles was already high in normoxia (almost 5 foci per cell), suggesting that cells were already stressed by the gapmer treatment, before being submitted to hypoxia. Our hypothesis to explain the effect of Neat1 depletion in normoxia is indeed that cells are stressed by the gapmer treatment. Regarding eIF2a, we performed the experiment (Figure 2—figure supplement 4): eIF2a is phosphorylated upon Neat1_2 gapmer treatment, but not using the control gapmer. This could explain the small difference of IRES activity between normoxia and hypoxia generated by the knock-down.

1. While NEAT1 abundance has an effect on IRES, the NEAT-ITAFs do not. The authors should check effects of DDX17, Eno3 and Hspa2 that were enriched in the cytoplasm during hypoxia (Figure 4E).

We chose to test only the knock-down of DDX17 (due to the difficulty to obtain the HL-1 culture medium with the COVID19 pandemic). As mentioned above, we observed an upregulation of the LucF/LucR ratio. We prefer no to publish this in the present paper because another study is being performed on the

role of several helicases and these data will be in our next publication.

2. All figures displaying FISH results are impossible to inspect. They need to be bigger.

We have improved the presentation of the FISH results.

3. Depletion of NEAT1 or NEAT1-ITAFs has always more effect on endogenous FGF1 protein abundance that on IRES activity. Is the half-life of FGF1 affected during the siRNA and gapmer treatments?

The half-life of FGF1 has been analyzed and does not vary upon gapmer treatment not with siRNA treatment (Figure 2—figure supplements 5-6 and Figure 4—figure supplements 3-4). It superior to 24 hr either without treatment, or with gapmer treatment, or with siRNA treatment (control and target-specific). A hypothesis to explain a more important effect on endogenous FGF1 mRNA than bicistronic mRNA translation may be that the IRES of the endogenous FGF1 mRNA is recruited more efficiently to the paraspeckle than the IRES present in the bicistronic mRNA, whose structure may be altered due to the presence of the upstream cistron. However, if one looks at the Western quantification (normalized to total proteins), the difference is not so big: upon Neat1 depletion endogenous FGF1 protein is 34%, while IRES activity is 50% (Figure 2C and 2D). Upon p54 or PSPC1 depletion, endogenous FGF1 is 61% or 82% while IRES activity is 70% or 80%, respectively (Figure 4E and 4F).

[Editors' note: further revisions were suggested prior to acceptance, as described below.]

The manuscript has been improved but there are some remaining issues that need to be addressed, as outlined below:As detailed by referees #1 and 2 there are omissions in the text of the authors' responses listed in the rebuttal letter. Most importantly, since many of the results are correlative the conclusions are overstated. As requested by referee # 1, since the authors do not show a direct association between Neat1 and the ITAFs investigated herein, the authors ought to change or remove Figure 4A which claims that Neat1 and p54nrb interact with the FGF1 IRES.Reviewer #1 (Recommendations for the authors):The authors have provided new information which addresses some of my previous concerns. However, my main concern remains in that there is much correlative data from which direct cause-effect relationships are inferred.

We thank you for your detailed analysis and we agree that we often present correlative data, because we were not able to demonstrate the direct cause-effect, but we provide a set of convincing arguments to strengthen our hypothesis. We have modified our conclusions when requested.

The authors indicate that there was a decrease in the number of paraspeckles in cells in which Neat1 was knocked down, as well as a 2-fold reduction in IRES activity. However, in new data presented, the authors show that eIF2alpha is phosphorylated upon Neat1_2 KO (Figure 2, Suppl 4) and in their rebuttal write that this could explain the small difference of IRES activity between normoxia and hypoxia generated by the KO. I, therefore, don't understand how the authors conclude on line 232 of the MS that the Neat1 knockdown results suggest "that Neat1 regulates FGF1 mRNA translation, directly or indirectly". The same comments holds true for the data regarding changes in the polysomal mRNA context following Neat1 knockdown (Figure 8). Are these due to elevated eIF2 α phosphorylation and not necessarily due to the absence of Neat1

This comment is indeed justified. We have deeply analyzed this issue from our data and we can provide a clear answer (added in the text page 5):

The IRES activity corresponds to the ratio LucF/LucR and provides a value normalized to LucR activity. If we look to the *Renilla* luciferase activity alone, reflecting the first cistron expression, we see that the slight increase in eIF2a phosphorylation is not sufficient to block global translation. We have added a small table in Supplementary file 3, page 2, showing the ratios of LucR activities with *Neat1* gapmer versus control gapmer: it shows that LucR expression is not silenced by *Neat1* depletion. Furthermore, we have shown in our previous report that the FGF1 IRES activity (as well as all IRES activities tested in our studies) increases in hypoxia in conditions of strong eIF2a phosphorylation (Hantelys et al., 2019). We have also mentioned this observation in page 4. This argument is the most important demonstration that the change in IRES activity is not due to eIF2a phosphorylation. The paper by Hantelys et al., 2019, published in *eLife* was performed in exactly the same conditions, with the same cells!.

The conclusion has been modified as following page 5:

“All these arguments indicate that the significant decrease of FGF1 IRES activity and of endogenous FGF1 expression observed in Figure 2 cannot result from eIF2a phosphorylation or decrease in FGF1 half-life, and probably results from Neat1 depletion. This suggests that *Neat1* might regulate *FGF1* mRNA translation, directly or indirectly.”

The text page 8 commenting Figure 8 has also been modified as following:

“As eIF2a phosphorylation was slightly increased in these conditions (Figure 2, figure supplement 4), we cannot completely rule out that it could affect the expression of certain mRNAs, despite the absence of inhibition of global translation. However, the insensitivity of many IRESs to eIF2a phosphorylation shown previously suggests that the present data result from an effect of *Neat1*, particularly on translation of IRES-containing mRNAs, while the two isoforms may have distinct effects (Hantelys et al., 2019).”

The authors use smiFISH to probe the localization of Neat 1 and FGF1 IRES and find a ~ 2-fold increase in overlapping signals for the FGF1 IRES and Neat1 in hypoxic versus normal cells. (Figure 3B). I'm not sure how this led to the "hypothesis that the paraspeckle might be involved in the control of IRES-dependent translation." For this conclusion to be made, wouldn't one have to show altered IRES activity after versus before having encountered the Neat1 speckles?

The conclusion of the paragraph commenting Figure 3 page 5 is:

“These data suggested that the IRES-containing mRNA is recruited into paraspeckles during hypoxia. This colocalization study, coupled to the functional study showing the involvement of all the major paraspeckle component in the IRES-dependent translation led us to the final hypothesis of our paper of a paraspeckle involvement. It is presented as a hypothesis and not as a conclusion.”

Since the authors do not show a direct association between Neat1 and the ITAFs investigated herein, I would ask that they change or remove Figure 4A which appears to show Neat1 and p54nrb interacting with the FGF1 IRES.

Done: the IRES-containing mRNA has been removed from the drawing.